# On the Convergence of A Class of Adam-Type Algorithms for Non-Convex Optimization

**Xiangyi Chen**[†], **Sijia Liu**[‡], **Ruoyu Sun**[§], **Mingyi Hong**[†]

[†]ECE, University of Minnesota - Twin Cities
[‡]MIT-IBM Watson AI Lab, IBM Research
[§]ISE, University of Illinois at Urbana-Champaign
[†]`{chen5719,mhong}@umn.edu`, [‡]`sijia.liu@ibm.com`, [§]`ruoyus@illinois.edu`

## Abstract

This paper studies a class of adaptive gradient based momentum algorithms that update the search directions and learning rates simultaneously using past gradients. This class, which we refer to as the "Adam-type", includes the popular algorithms such as Adam (Kingma & Ba, 2014) , AMSGrad (Reddi et al., 2018) , AdaGrad (Duchi et al., 2011). Despite their popularity in training deep neural networks (DNNs), the convergence of these algorithms for solving non-convex problems remains an *open* question.

In this paper, we develop an analysis framework and a set of mild sufficient conditions that guarantee the convergence of the Adam-type methods, with a convergence rate of order $O(\log T/\sqrt{T})$ for non-convex stochastic optimization. Our convergence analysis applies to a new algorithm called AdaFom (AdaGrad with First Order Momentum). We show that the conditions are essential, by identifying concrete examples in which violating the conditions makes an algorithm diverge. Besides providing one of the first comprehensive analysis for Adam-type methods in the non-convex setting, our results can also help the practitioners to easily monitor the progress of algorithms and determine their convergence behavior.

## 1 Introduction

First-order optimization has witnessed tremendous progress in the last decade, especially to solve machine learning problems (Bottou et al., 2018). Almost every first-order method obeys the following generic form (Boyd & Vandenberghe, 2004), $\mathbf{x}_{t+1} = \mathbf{x}_t - \alpha_t \mathbf{\Delta}_t$, where $\mathbf{x}_t$ denotes the solution updated at the $t$th iteration for $t = 1, 2, \ldots, T$, $T$ is the number of iterations, $\mathbf{\Delta}_t$ is a certain (approximate) descent direction, and $\alpha_t > 0$ is some learning rate. The most well-known first-order algorithms are gradient descent (GD) for deterministic optimization (Nesterov, 2013; Cartis et al., 2010) and stochastic gradient descent (SGD) for stochastic optimization (Zinkevich, 2003; Ghadimi & Lan, 2013), where the former determines $\mathbf{\Delta}_t$ using the full (batch) gradient of an objective function, and the latter uses a simpler but more computationally-efficient stochastic (unbiased) gradient estimate.

Recent works have proposed a variety of accelerated versions of GD and SGD (Nesterov, 2013). These achievements fall into three categories: a) *momentum methods* (Nesterov, 1983; Polyak, 1964; Ghadimi et al., 2015) which carefully design the descent direction $\mathbf{\Delta}_t$; b) *adaptive learning rate methods* (Becker et al., 1988; Duchi et al., 2011; Zeiler, 2012; Dauphin et al., 2015) which determine good learning rates $\alpha_t$, and c) *adaptive gradient methods* that enjoy dual advantages of a) and b). In particular, Adam (Kingma & Ba, 2014), belonging to the third type of methods, has become extremely popular to solve deep learning problems, e.g., to train deep neural networks. Despite its superior performance in practice, theoretical investigation of Adam-like methods for *non-convex* optimization is still missing.

Very recently, the work (Reddi et al., 2018) pointed out the convergence issues of Adam even in the convex setting, and proposed AMSGrad, a corrected version of Adam. Although AMSGrad has made a positive step towards understanding the theoretical behavior of adaptive gradient methods, the convergence analysis of (Reddi et al., 2018) was still very restrictive because it only works for convex problems, despite the fact that the most successful applications are for non-convex problems. Apparently, there still exists a large gap between theory and practice. To the best of our knowledge,

the question that whether adaptive gradient methods such as Adam, AMSGrad, AdaGrad converge for non-convex problems is still open in theory.

After the non-convergence issue of Adam has been raised in (Reddi et al., 2018), there have been a few recent works on proposing new variants of Adam-type algorithms. In the convex setting, reference (Huang et al., 2018) proposed to stabilize the coordinate-wise weighting factor to ensure convergence. Reference (Chen & Gu, 2018) developed an algorithm that changes the coordinate-wise weighting factor to achieve better generalization performance. Concurrent with this work, several works are trying to understand performance of Adam in non-convex optimization problems. Reference (Basu et al., 2018) provided convergence rate of original Adam and RMSprop under full-batch (deterministic) setting, and (Ward et al., 2018) proved convergence rate of a modified version of AdaGrad where coordinate-wise weighting is removed. Furthermore, the work (Zhou et al., 2018) provided convergence results for AMSGrad that exhibit a tight dependency on problem dimension compared to (Reddi et al., 2018). The works (Zou & Shen, 2018) and (Li & Orabona, 2018) proved that both AdaGrad and its variant (AdaFom) converge to a stationary point with a high probability. The aforementioned works are independent of ours. In particular, our analysis is not only more comprehensive (it covers the analysis of a large family of algorithms in a single framework), but more importantly, it provides insights on how oscillation of stepsizes can affect the convergence rate.

**Contributions**  Our work aims to build the theory to understand the behavior for a *generic* class of adaptive gradient methods for non-convex optimization. In particular, we provide mild sufficient conditions that guarantee the convergence for the Adam-type methods. We summarize our contribution as follows.

• **(Generality)** We consider a class of generalized Adam, referred to as the "Adam-type", and we show for the first time that under suitable conditions about the stepsizes and algorithm parameters, this class of algorithms all converge to first-order stationary solutions of the non-convex problem, with $O(\log T/\sqrt{T})$ convergence rate. This class includes the recently proposed AMSGrad (Reddi et al., 2018), AdaGrad (Duchi et al., 2011), and stochastic heavy-ball methods as well as two new algorithms explained below.

> • **(AdaFom)** Adam adds momentum to both the first and the second moment estimate, but this leads to possible divergence (Reddi et al., 2018). We show that the divergence issue can actually be fixed by a simple variant which adds momentum to only the first moment estimate while using the same second moment estimate as that of AdaGrad, which we call AdaFom (AdaGrad with First Order Moment).
>
> • **(Constant Momemtum)** Our convergence analysis is applicable to the *constant* momentum parameter setting for AMSGrad and AdaFom. The divergence example of Adam given in (Reddi et al., 2018) is for constant momentum parameter, but the convergence analysis of AMSGrad in (Reddi et al., 2018) is for diminishing momentum parameter. This discrepancy leads to a question whether the convergence of AMSGrad is due to the algorithm form or due to the momentum parameter choice – we show that the constant-momentum version of AMSGrad indeed converges, thus excluding the latter possibility.

• **(Practicality)** The sufficient conditions we derive are simple and easy to check in practice. They can be used to either certify the convergence of a given algorithm for a class of problem instances, or to track the progress and behavior of a particular realization of an algorithm.

• **(Tightness and Insight)** We show the conditions are essential and "tight", in the sense that violating them can make an algorithm diverge. Importantly, our conditions provide insights on how oscillation of a so-called "effective stepsize" (that we define later) can affect  the convergence rate of the class of algorithms. We also provide interpretations of the convergence conditions  to illustrate why under some circumstances, certain Adam-type algorithms can outperform SGD.

**Notations**  We use $z = x/y$ to denote element-wise division if $x$ and $y$ are both vectors of size $d$; $x \odot y$ is element-wise product, $x^2$ is element-wise square if $x$ is a vector, $\sqrt{x}$ is element-wise square root if $x$ is a vector, $(x)_j$ denotes $j$th coordinate of $x$, $\|x\|$ is $\|x\|_2$ if not otherwise specified. We use $[N]$ to denote the set $\{1, \cdots, N\}$, and use $O(\cdot), o(\cdot), \Omega(\cdot), \omega(\cdot)$ as standard asymptotic notations.

## 2   PRELIMINARIES AND ADAM-TYPE ALGORITHMS

Stochastic optimization is a popular framework for analyzing algorithms in machine learning due to the popularity of mini-batch gradient evaluation. We consider the following generic problem where

we are minimizing a function $f$, expressed in the expectation form as follows

$$\min_{x \in \mathbb{R}^d} f(x) = \mathbb{E}_\xi[f(x; \xi)], \tag{1}$$

where $\xi$ is a certain random variable representing randomly selected data sample or random noise.

In a generic first-order optimization algorithm, at a given time $t$ we have access to an unbiased noisy gradient $g_t$ of $f(x)$, evaluated at the current iterate $x_t$. The noisy gradient is assumed to be bounded and the noise on the gradient at different time $t$ is assumed to be independent. An important assumption that we will make throughout this paper is that the function $f(x)$ is continuously differentiable and has Lipschitz continuous gradient, but could otherwise be a *non-convex* function. The non-convex assumption represents a major departure from the convexity that has been assumed in recent papers for analyzing Adam-type methods, such as (Kingma & Ba, 2014) and (Reddi et al., 2018).

Our work focuses on the generic form of exponentially weighted stochastic gradient descent method presented in Algorithm 1, for which we name as *generalized Adam* due to its resemblance to the original Adam algorithm and many of its variants.

---

**Algorithm 1. Generalized Adam**

**S0.** Initialize $m_0 = 0$ and $x_1$
For $t = 1, \cdots, T$, do
    **S1.** $m_t = \beta_{1,t}m_{t-1} + (1 - \beta_{1,t})g_t$
    **S2.** $\hat{v}_t = h_t(g_1, g_2, ..., g_t)$
    **S3.** $x_{t+1} = x_t - \alpha_t m_t / \sqrt{\hat{v}_t}$
End

---

In Algorithm 1, $\alpha_t$ is the step size at time $t$, $\beta_{1,t} > 0$ is a sequence of problem parameters, $m_t \in \mathbb{R}^d$ denotes some (exponentially weighted) gradient estimate, and $\hat{v}_t = h_t(g_1, g_2, ..., g_t) \in \mathbb{R}^d$ takes all the past gradients as input and returns a vector of dimension $d$, which is later used to inversely weight the gradient estimate $m_t$. And note that $m_t / \sqrt{\hat{v}_t} \in \mathbb{R}^d$ represents element-wise division. Throughout the paper, we will refer to the vector $\alpha_t / \sqrt{\hat{v}_t}$ as the *effective stepsize*.

We highlight that Algorithm 1 includes many well-known algorithms as special cases. We summarize some popular variants of the generalized Adam algorithm in Table 1.

**Table 1:** Variants of generalized Adam

| $\hat{v}_t$ $\diagdown$ $\beta_{1,t}$ | $\beta_{1,t} = 0$ | $\beta_{1,t} \le \beta_{1,t-1}$ $\beta_{1,t} \xrightarrow[t\to\infty]{} b \ge 0$ | $\beta_{1,t} = \beta_1$ |
|---|---|---|---|
| $\hat{v}_t = 1$ | SGD | N/A* | Heavy-ball method |
| $\hat{v}_t = \frac{1}{t}\sum_{i=1}^t g_i^2$ | AdaGrad | AdaFom | AdaFom |
| $v_t = \beta_2 v_{t-1} + (1 - \beta_2)g_t^2,$ $\hat{v}_t = \max(\hat{v}_{t-1}, v_t)$ | AMSGrad | AMSGrad | AMSGrad |
| $\hat{v}_t = \beta_2\hat{v}_{t-1} + (1 - \beta_2)g_t^2$ | RMSProp | N/A | Adam |

   * N/A stands for an informal algorithm that was not defined in literature.

We present some interesting findings for the algorithms presented in Table 1.

- Adam is often regarded as a "momentum version" of AdaGrad, but it is different from AdaFom which is also a momentum version of AdaGrad [1]. The difference lies in the form of $\hat{v}_t$. Intuitively, Adam adds momentum to both the first and second order moment estimate, while in AdaFom we only add momentum to the first moment estimate and use the same second moment estimate as AdaGrad. These two methods are related in the following way: if we let $\beta_2 = 1 - 1/t$ in the expression of $\hat{v}_t$ in Adam, we obtain AdaFom. We can view AdaFom as a variant of Adam with an increasing sequence of $\beta_2$, or view Adam as a variant of AdaFom with exponentially decaying weights of $g_t^2$. However, this small change has large impact on the convergence: we prove that AdaFom can always converge under standard

---

[1] AdaGrad with first order momentum is also studied in (Zou & Shen, 2018) which appeared online after our first version

assumptions (see Corollary 3.2) , while Adam is shown to possibly diverge (Reddi et al., 2018).

- The convergence of AMSGrad using a fast diminishing $\beta_{1,t}$ such that $\beta_{1,t} \leq \beta_{1,t-1}, \beta_{1,t} \xrightarrow[t \to \infty]{} b, b = 0$ in convex optimization was studied in (Reddi et al., 2018). However, the convergence of the version with constant $\beta_1$ or strictly positive $b$ and the version for non-convex setting are unexplored before our work. We notice that an independent work (Zhou et al., 2018) has also proved the convergence of AMSGrad with constant $\beta_1$.

It is also worth mentioning that Algorithm 1 can be applied to solve the popular "finite-sum" problems whose objective is a sum of $n$ individual cost functions. That is,

$$\min_{\mathbf{x} \in \mathbb{R}^d} \sum_{i=1}^n f_i(x) := f(x), \tag{2}$$

where each $f_i : \mathbb{R}^d \to \mathbb{R}$ is a smooth and possibly non-convex function. If at each time instance the index $i$ is chosen  uniformly randomly, then Algorithm 1 still applies, with $g_t = \nabla f_i(x_t)$. It can also be extended to a mini-batch case with $\mathbf{g}_t = \frac{1}{b} \sum_{i \in \mathcal{I}_t} \nabla f_i(\mathbf{x}_t)$, where $\mathcal{I}_t$ denotes the minibatch of size $b$ at time $t$. It is easy to show that $g_t$ is an unbiased estimator for $\nabla f(x)$.

In the remainder of this paper, we will analyze Algorithm 1 and provide sufficient conditions under which the algorithm  converges to first-order stationary solutions with sublinear rate. We will also discuss how our results can be applied to special cases of generalized Adam.

## 3   Convergence Analysis for Generalized Adam

The main technical challenge in analyzing the non-convex version of Adam-type algorithms is that the actually used update directions could no longer be unbiased estimates of the true gradients. Furthermore, an additional difficulty is introduced by the involved form of the adaptive learning rate. Therefore the biased gradients have to be carefully analyzed together with the use of the inverse of exponential moving average while adjusting the learning rate. The existing convex analysis (Reddi et al., 2018) does not apply to the non-convex scenario  we study for at least two reasons: first, non-convex optimization requires a different convergence criterion, given by stationarity rather than the global optimality; second, we consider *constant* momentum controlling parameter.

In the following, we formalize the assumptions required in our convergence analysis.

**Assumptions**

A1: $f$ is differentiable and has $L$-Lipschitz gradient, i.e. $\forall x, y, \|\nabla f(x) - \nabla f(y)\| \leq L\|x - y\|$. It is also lower bounded, i.e. $f(x^*) > -\infty$ where $x^*$ is an optimal solution.

A2: At time $t$, the algorithm can access a bounded noisy gradient and the true gradient is bounded, i.e. $\|\nabla f(x_t)\| \leq H, \quad \|g_t\| \leq H, \quad \forall t > 1$.

A3: The noisy gradient is unbiased and the noise is independent, i.e. $g_t = \nabla f(x_t) + \zeta_t, E[\zeta_t] = 0$ and $\zeta_i$ is independent of $\zeta_j$ if $i \neq j$.

Reference (Reddi et al., 2018) uses a similar (but slightly different) assumption as A2, i.e., the bounded elements of the gradient $\|g_t\|_\infty \leq a$ for some finite $a$. The bounded norm of $\nabla f(x_t)$ in A2 is equivalent to Lipschitz continuity of $f$ (when $f$ is differentiable) which is a commonly used condition in convergence analysis. This assumption is often satisfied in practice, for example it holds for the finite sum problem (2) when each $f_i$ has bounded gradient, and $g_t = \nabla f_i(x_t)$ where $i$ is sampled randomly. A3 is also standard in stochastic optimization for analyzing convergence.

Our main result shows that if the coordinate-wise weighting term $\sqrt{\hat{v}_t}$ in Algorithm 1 is properly chosen, we can ensure the global convergence as well as the sublinear convergence rate of the algorithm (to a first-order stationary solution). First, we characterize how the effective stepsize parameters $\alpha_t$ and $\hat{v}_t$ affect convergence of Adam-type algorithms.

**Theorem 3.1.** *Suppose that Assumptions A1-A3 are satisfied, $\beta_1$ is chosen such that $\beta_1 \geq \beta_{1,t}$, $\beta_{1,t} \in [0, 1)$ is non-increasing, and for some constant $G > 0$, $\left\| \alpha_t m_t / \sqrt{\hat{v}_t} \right\| \leq G, \; \forall \; t$. Then*

*Algorithm 1 yields*

$$E\left[\sum_{t=1}^{T}\alpha_t\langle\nabla f(x_t),\nabla f(x_t)/\sqrt{\hat{v}_t}\rangle\right] \tag{3}$$

$$\leq E\left[C_1\sum_{t=1}^{T}\left\|\alpha_t g_t/\sqrt{\hat{v}_t}\right\|^2 + C_2\sum_{t=2}^{T}\left\|\frac{\alpha_t}{\sqrt{\hat{v}_t}}-\frac{\alpha_{t-1}}{\sqrt{\hat{v}_{t-1}}}\right\|_1 + C_3\sum_{t=2}^{T-1}\left\|\frac{\alpha_t}{\sqrt{\hat{v}_t}}-\frac{\alpha_{t-1}}{\sqrt{\hat{v}_{t-1}}}\right\|^2\right] + C_4$$

*where $C_1, C_2, C_3$ are constants independent of $d$ and $T$, $C_4$ is a constant independent of $T$, the expectation is taken with respect to all the randomness corresponding to $\{g_t\}$.*

*Further, let $\gamma_t := \min_{j\in[d]}\min_{\{g_i\}_{i=1}^t} \alpha_t/(\sqrt{\hat{v}_t})_j$ denote the minimum possible value of effective stepsize at time $t$ over all possible coordinate and past gradients $\{g_i\}_{i=1}^t$. Then the convergence rate of Algorithm 1 is given by*

$$\min_{t\in[T]} E\left[\|\nabla f(x_t)\|^2\right] = O\left(\frac{s_1(T)}{s_2(T)}\right), \tag{4}$$

*where $s_1(T)$ is defined through the upper bound of RHS of (3), namely, $O(s_1(T))$, and $\sum_{t=1}^{T}\gamma_t = \Omega(s_2(T))$.*

**Proof**: See Appendix 6.2. ∎

In Theorem 3.1, $\|\alpha_t m_t/\sqrt{\hat{v}_t}\| \leq G$ is a mild condition. Roughly speaking, it implies that the change of $x_t$ at each each iteration should be finite. As will be evident later, with $\|g_t\| \leq H$, the condition $\|\alpha_t m_t/\sqrt{\hat{v}_t}\| \leq G$ is automatically satisfied for both AdaGrad and AMSGrad. Besides, instead of bounding the minimum norm of $\nabla f$ in (4), we can also apply a probabilistic output (e.g., select an output $\mathbf{x}_R$ with probability $p(R = t) = \frac{\gamma_t}{\sum_{t=1}^T \gamma_t}$) to bound $E[\|\nabla f(x_R)\|^2]$ (Ghadimi & Lan, 2013; Lei et al., 2017). It is worth mentioning that a small number $\epsilon$ could be added to $\hat{v}_t$ for ensuring the numerical stability. In this case, our Theorem 3.1 still holds given the fact the resulting algorithm is still a special case of Algorithm 1. Accordingly, our convergence results for AMSGrad and AdaFom that will be derived later also hold as $\|\alpha_t m_t/(\sqrt{\hat{v}_t}+\epsilon)\| \leq \|\alpha_t m_t/\sqrt{\hat{v}_t}\| \leq G$ when $\epsilon$ is added to $\hat{v}_t$. We will provide a detailed explanation of Theorem 3.1 in Section 3.1.

Theorem 3.1 implies a sufficient condition that guarantees convergence of the Adam-type methods: $s_1(T)$ grows slower than $s_2(T)$. We will show in Section 3.2 that the rate $s_1(T)$ can be dominated by different terms in different cases, i.e. the non-constant quantities Term A and B below

$$E\left[\underbrace{\sum_{t=1}^{T}\left\|\alpha_t g_t/\sqrt{\hat{v}_t}\right\|^2}_{\text{Term A}} + \underbrace{\sum_{t=2}^{T}\left\|\frac{\alpha_t}{\sqrt{\hat{v}_t}}-\frac{\alpha_{t-1}}{\sqrt{\hat{v}_{t-1}}}\right\|_1}_{\text{Term B}} + \sum_{t=2}^{T-1}\left\|\frac{\alpha_t}{\sqrt{\hat{v}_t}}-\frac{\alpha_{t-1}}{\sqrt{\hat{v}_{t-1}}}\right\|^2\right] = O(s_1(T)), \tag{5}$$

where the growth of third term at LHS of (5) can be directly related to growth of Term B via the relationship between $\ell_1$ and $\ell_2$ norm or upper boundedness of $(\alpha_t/\sqrt{\hat{v}_t})_j$.

### 3.1 EXPLANATION OF CONVERGENCE CONDITIONS

From (4) in Theorem 3.1, it is evident that $s_1(T) = o(s_2(T))$ can ensure proper convergence of the algorithm. This requirement has some important implications, which we discuss below.

• **(The Bounds for $s_1(T)$ and $s_2(T)$)** First, the requirement that $s_1(T) = o(s_2(T))$ implies that $E[\sum_{t=1}^{T}\|\alpha_t g_t/\sqrt{\hat{v}_t}\|^2] = o(\sum_{t=1}^{T}\gamma_t)$. This is a common condition generalized from SGD. Term A in (5) is a generalization of the term $\sum_{t=1}^{T}\|a_t g_t\|^2$ for SGD (where $\{\alpha_t\}$ is the stepsize sequence for SGD), and it quantifies possible increase in the objective function brought by higher order curvature. The term $\sum_{t=1}^{T}\gamma_t$ is the lower bound on the summation of effective stepsizes, which reduces to $\sum_{t=1}^{T}\alpha_t$ when Algorithm 1 is simplified to SGD.
• **(Oscillation of Effective Stepsizes)** Term B in (5) characterizes the *oscillation of effective stepsizes* $\alpha_t/\sqrt{\hat{v}_t}$. In our analysis such an oscillation term upper bounds the expected possible ascent in objective induced by skewed update direction $g_t/\sqrt{\hat{v}_t}$ ("skewed" in the sense that $E[g_t/\sqrt{\hat{v}_t}]$ is not parallel with $\nabla f(x_t)$), therefore it cannot be too large. Bounding this term is critical, and to demonstrate this fact, in Section 3.2.2 we show that large oscillation can result in non-convergence of Adam for even simple unconstrained non-convex problems.
• **(Advantage of Adaptive Gradient).** One possible benefit of adaptive gradient methods can be seen from Term A. When this term dominates the convergence speed in Theorem 3.1, it is possible

that proper design of $\hat{v}_t$ can help reduce this quantity compared with SGD (An example is provided in Appendix 6.1.1 to further illustrate this fact.) in certain cases. Intuitively, adaptive gradient methods like AMSGrad can provide a flexible choice of stepsizes, since $\hat{v}_t$ can have a *normalization effect* to reduce oscillation and overshoot introduced by large stepsizes. At the same time, flexibility of stepsizes makes the hyperparameter tuning of an algorithm easier in practice.

## 3.2 Tightness of the rate bound (4)

In the next, we show our bound (4) is tight in the sense that there exist problems satisfying Assumption 1 such that certain algorithms belonging to the class of Algorithm 1 can diverge due to the high growth rate of Term A or Term B.

### 3.2.1 Non-convergence of SGD and Adam due to effect of Term A

We demonstrate the importance of Term A in this subsection. Consider a simple one-dimensional optimization problem $\min_x f(x)$, with $f(x) = 100x^2$ if $|x| <= b$, and $f(x) = 200b|x| - 100b^2$ if $|x| > b$, where $b = 10$. In Figure 1, we show the growth rate of different terms given in Theorem 3.1, where $\alpha_0 \triangleq 0$, $\alpha_t = 0.01$ for $t \geq 1$, and $\beta_{1,t} = 0, \beta_{2,t} = 0.9$ for both Adam and AMSGrad. We observe that both SGD and Adam are not converging to a stationary solution ($x = 0$), which is because $\sum_{t=1}^{T} \left\| \alpha_t g_t / \sqrt{\hat{v}_t} \right\|^2$ grows with the same rate as accumulation of effective stepsizes as shown in the figure. Actually, SGD only converges when $\alpha_t < 0.01$ and our theory provides an perspective of why SGD diverges when $\alpha_t \geq 0.01$. In the example, Adam is also not converging to 0 due to Term A. From our observation, Adam oscillates for any constant stepsize within $[10^{-4}, 0.1]$ for this problem and Term A always ends up growing as fast as accumulation of effective stepsizes, which implies Adam only converges with diminishing stepsizes even in non-stochastic optimization. In contrast to SGD and Adam, AMSGrad converges in this case since both Term A and Term B grow slower than accumulation of effective stepsizes. For AMSGrad and Adam, $\hat{v}_t$ has a strong normalization effect and it allows the algorithm to use a larger range of $\alpha_t$. The practical benefit of this flexible choice of stepsizes is easier hyperparameter tuning, which is consistent with the impression of practitioners about the original Adam. We present more experimental results in Appendix 6.1.1 accompanied with more detailed discussions.

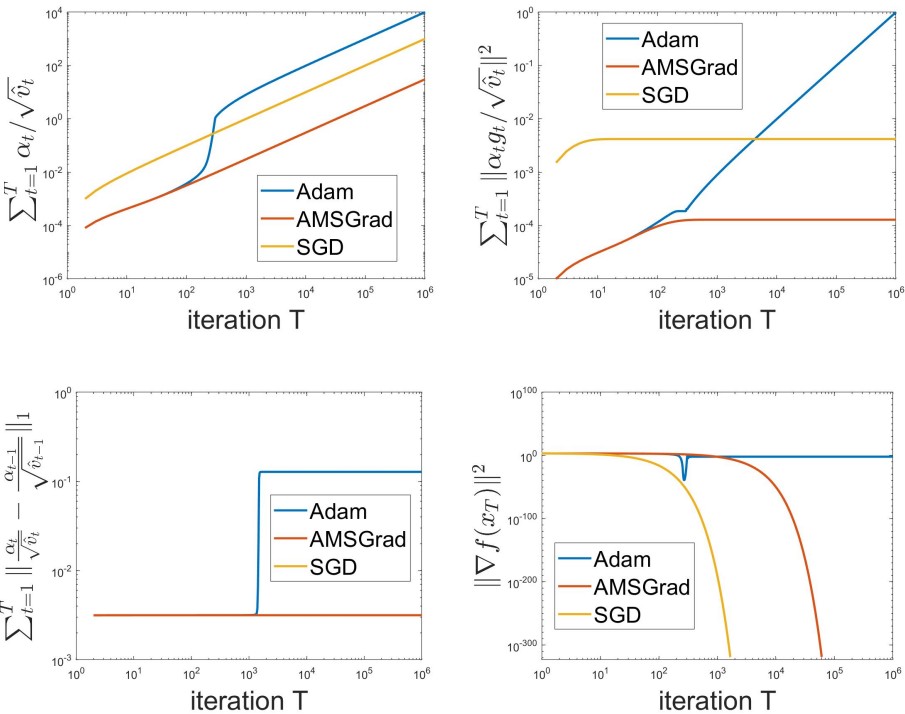

**Figure 1:** A toy example to illustrate effect of Term A on Adam, AMSGrad, and SGD.

### 3.2.2   NON-CONVERGENCE OF ADAM DUE TO EFFECT OF TERM B

Next, we use an example to demonstrate the importance of the Term B for the convergence of Adam-type algorithms.

Consider optimization problem $\min_x f(x) = \sum_{i=1}^{11} f_i(x)$ where

$$f_i(x) = \begin{cases} \mathbb{I}[i=1]5.5x^2 + \mathbb{I}[i \neq 1](-0.5x^2), & \text{if } |x| \leq 1 \\ \mathbb{I}[i=1](11|x|-5.5) + \mathbb{I}[i \neq 1](-|x|+0.5), & \text{if } |x| > 1 \end{cases} \tag{6}$$

and $\mathbb{I}[1=1] = 1, \mathbb{I}[1 \neq 1] = 0$. It is easy to verify that the only point with $\nabla f(x) = 0$ is $x = 0$. The problem satisfies the assumptions in Theorem 3.1 as the stochastic gradient $g_t = \nabla f_i(x_t)$ is sampled uniformly for $i \in [11]$. We now use the AMSGrad and Adam to optimize $x$, and the results are given in Figure 2, where we set $\alpha_t = 1$, $\beta_{1,t} = 0$, and $\beta_{2,t} = 0.1$. We observe that $\sum_{t=1}^{T} \|\alpha_t/\sqrt{\hat{v}_t} - \alpha_{t-1}/\sqrt{\hat{v}_{t-1}}\|_1$ in Term B grows with the same rate as $\sum_{t=1}^{T} \alpha_t/\sqrt{\hat{v}_t}$ for Adam, where we recall that $\sum_{t=1}^{T} \alpha_t/\sqrt{\hat{v}_t}$ is an upper bound of $\sum_{t=1}^{T} \gamma_t$ in Theorem 3.1. As a result, we obtain $O(s_1(T)/s_2(T)) \neq o(1)$ in (4), implying the non-convergence of Adam. Our theoretical analysis matches the empirical results in Figure 2. In contrast, AMSGrad converges in Figure 2 because of its smaller oscillation in effective stepsizes, associated with Term B. We finally remark that the importance of the quantity $\sum_{t=1}^{T} \|\alpha_t/\sqrt{\hat{v}_t} - \alpha_{t-1}/\sqrt{\hat{v}_{t-1}}\|_1$ is also noticed by (Huang et al., 2018). However, they did not analyze its effect on convergence, and their theory is only for convex optimization.

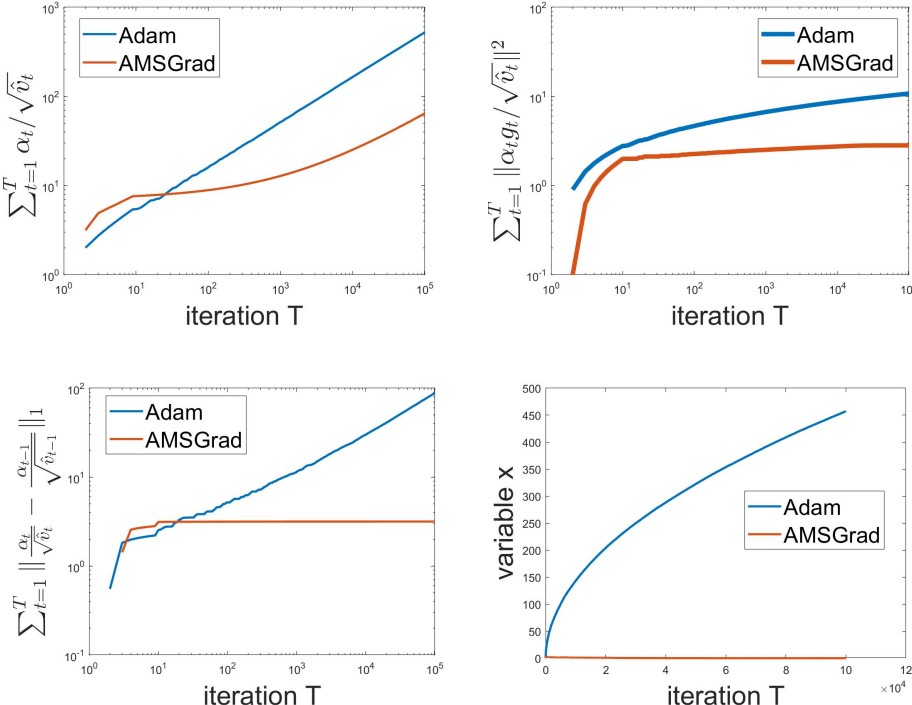

**Figure 2:** A toy example to illustrate effect of Term B on Adam and AMSGrad.

### 3.3   CONVERGENCE OF AMSGRAD AND ADAFOM

Theorem 3.1 provides a general approach for the design of the weighting sequence $\{\hat{v}_t\}$ and the convergence analysis of Adam-type algorithms. For example, SGD specified by Table 1 with stepsizes $\alpha_t = 1/\sqrt{t}$ yields $O(\log T/\sqrt{T})$ convergence speed by Theorem 3.1. Moreover, the explanation on the non-convergence of Adam in (Reddi et al., 2018) is consistent with our analysis in Section 3.2. That is, Term B in (5) can grow as fast as $s_2(T)$ so that $s_1(T)/s_2(T)$ becomes a constant. Further, we notice that Term A in (5) can also make Adam diverge which is unnoticed before. Aside from checking convergence of an algorithm, Theorem 3.1 can also provide convergence rates of AdaGrad and AMSGrad, which will be given as corollaries later.

Our proposed convergence rate of AMSGrad matches the result in (Reddi et al., 2018) for stochastic convex optimization. However, the analysis of AMSGrad in (Reddi et al., 2018) is constrained to diminishing momentum controlling parameter $\beta_{1,t}$. Instead, our analysis is applicable to the more popular *constant* momentum parameter, leading to a more general *non-increasing* parameter setting.

In Corollary 3.1 and Corollary 3.2, we derive the convergence rates of AMSGrad (Algorithm 3 in Appendix 6.2.3) and AdaFom (Algorithm 4 in Appendix 6.2.4), respectively. Note that AdaFom is more general than AdaGrad since when $\beta_{1,t} = 0$, AdaFom becomes AdaGrad.

**Corollary 3.1.** *Assume $\exists c > 0$ such that $|(g_1)_i| \geq c, \forall i \in [d]$, for AMSGrad (Algorithm 3 in Appendix 6.2.3) with $\beta_{1,t} \leq \beta_1 \in [0,1)$ and $\beta_{1,t}$ is non-increasing, $\alpha_t = 1/\sqrt{t}$, we have for any $T$,*

$$\min_{t \in [T]} E\left[\|f(x_t)\|^2\right] \leq \frac{1}{\sqrt{T}}\left(Q_1 + Q_2 \log T\right) \tag{7}$$

*where $Q_1$ and $Q_2$ are two constants independent of $T$.*

**Proof**: See Appendix 6.2.3. ∎

**Corollary 3.2.** *Assume $\exists c > 0$ such that $|(g_1)_i| \geq c, \forall i \in [d]$, for AdaFom (Algorithm 4 in Appendix 6.2.4) with $\beta_{1,t} \leq \beta_1 \in [0,1)$ and $\beta_{1,t}$ is non-increasing, $\alpha_t = 1/\sqrt{t}$, we have for any $T$,*

$$\min_{t \in [T]} E\left[\|f(x_t)\|^2\right] \leq \frac{1}{\sqrt{T}}(Q_1' + Q_2' \log T) \tag{8}$$

*where $Q_1'$ and $Q_2'$ are two constants independent of $T$.*

**Proof**: See Appendix 6.2.4. ∎

The assumption $|(g_1)_i| \geq c, \forall i$ is a mild assumption and it is used to ensure $\hat{v}_1 \geq r$ for some constant $r$. It is also usually needed in practice for numerical stability (for AMSGrad and AdaGrad, if $(g_1)_i = 0$ for some $i$, division by 0 error may happen at the first iteration). In some implementations, to avoid numerical instability, the update rule of algorithms like Adam, AMSGrad, and AdaGrad take the form of $x_{t+1} = x_t - \alpha_t m_t/(\sqrt{\hat{v}_t} + \epsilon)$ with $\epsilon$ being a positive number. These modified algorithms still fall into the framework of Algorithm 1 since $\epsilon$ can be incorporated into the definition of $\hat{v}_t$. Meanwhile, our convergence proof for Corollary 3.1 and Corollary 3.2 can go through without assuming $|(g_1)_i| \geq c, \forall i$ because $\sqrt{\hat{v}_t} \geq \epsilon$. In addition, $\epsilon$ can affect the worst case convergence rate by a constant factor in the analysis.

We remark that the derived convergence rate of AMSGrad and AdaFom involves an additional $\log T$ factor compared to the fastest rate of first order methods $(1/\sqrt{T})$. However, such a slowdown can be mitigated by choosing an appropriate stepsize. To be specific, the $\log T$ factor for AMSGrad would be eliminated when we adopt a constant rather than diminishing stepsize, e.g., $\alpha_t = 1/\sqrt{T}$. It is also worth mentioning that our theoretical analysis focuses on the convergence rate of adaptive methods in the worst case for nonconvex optimization. Thus, a sharper convergence analysis that can quantify the benefits of adaptive methods still remains an open question in theory.

## 4 Empirical performance of Adam-type algorithms on MNIST

In this section, we compare the empirical performance of Adam-type algorithms, including AMSGrad, Adam, AdaFom and AdaGrad, on training two convolutional neural networks (CNNs). In the first example, we train a CNN of 3 convolutional layers and 2 fully-connected layers on MNIST. In the second example, we train a CIFARNET on CIFAR-10. We refer readers to Appendix 6.1.2 for more details on the network model and the parameter setting.

In Figure 3, we present the training loss and the classification accuracy of Adam-type algorithms versus the number of iterations. As we can see, AMSGrad performs quite similarly to Adam which confirms the result in (Reddi et al., 2018). The performance of AdaGrad is worse than other algorithms, because of the lack of momentum and/or the significantly different choice of $\hat{v}_t$. We also observe that the performance of AdaFom lies between AMSGrad/Adam and AdaGrad. This is not surprising, since AdaFom can be regarded as a momentum version of AdaGrad but uses a simpler adaptive learning rate (independent on $\beta_2$) compared to AMSGrad/Adam. In Figure 4, we consider to train a larger network (CIFARNET) on CIFAR-10. As we can see, Adam and AMSGrad perform similarly and yield the best accuracy. AdaFom outperforms AdaGrad in both training and testing, which agrees with the results obtained in the MNIST experiment.

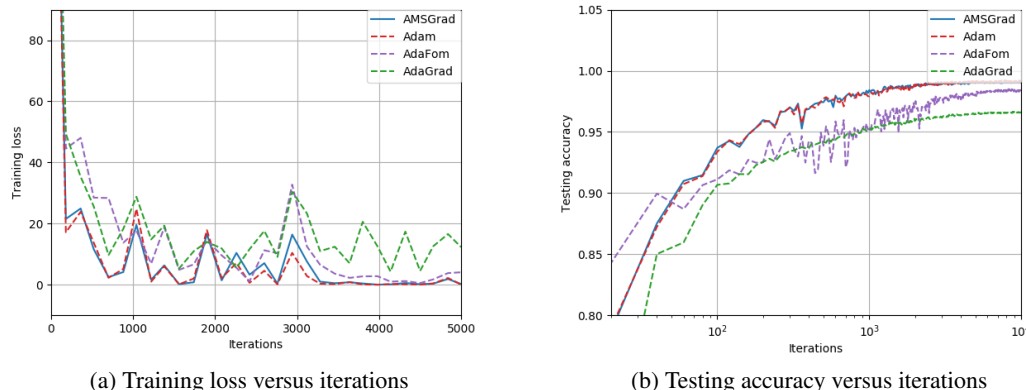

(a) Training loss versus iterations

(b) Testing accuracy versus iterations

**Figure 3:** Comparison of AMSGrad, Adam, AdaFom and AdaGrad under MNIST in training loss and testing accuracy.

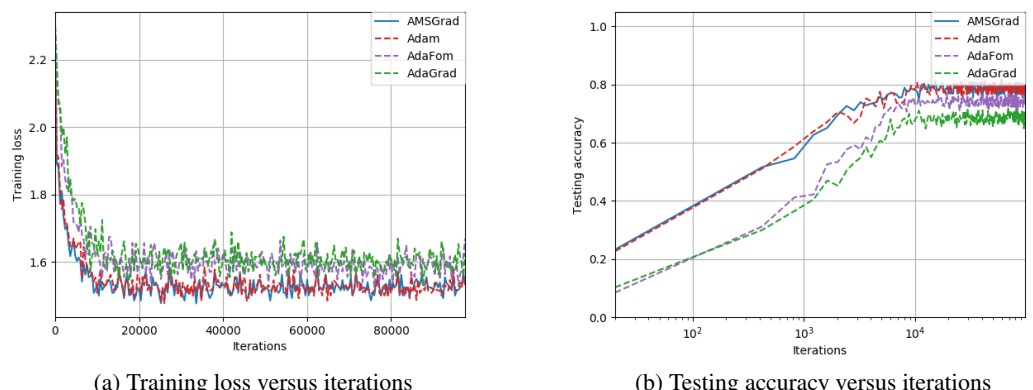

(a) Training loss versus iterations

(b) Testing accuracy versus iterations

**Figure 4:** Comparison of AMSGrad, Adam, AdaFom and AdaGrad under CIFAR in training loss and testing accuracy.

## 5 CONCLUSION AND DISCUSSION

We provided some mild conditions to ensure convergence of a class of Adam-type algorithms, which includes Adam, AMSGrad, AdaGrad, AdaFom, SGD, SGD with momentum as special cases. Apart from providing general convergence guarantees for algorithms, our conditions can also be checked in practice to monitor empirical convergence. To the best of our knowledge, the convergence of Adam-type algorithm for non-convex problems was unknown before. We also provide insights on how oscillation of effective stepsizes can affect convergence rate for the class of algorithms which could be beneficial for the design of future algorithms. This paper focuses on unconstrained non-convex optimization problems, and one future direction is to study a more general setting of constrained non-convex optimization.

### ACKNOWLEDGMENTS

This work was supported by the MIT-IBM Watson AI Lab. Mingyi Hong and Xiangyi Chen are supported partly by an NSF grant CMMI-1727757,and by an AFOSR grant 15RT0767.

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

## 6 Appendix

### 6.1 Related Work

*Momentum methods* take into account the history of first-order information (Nesterov, 2013; 1983; Nemirovskii et al., 1983; Ghadimi & Lan, 2016; Polyak, 1964; Ghadimi et al., 2015; Ochs et al., 2015; Yang et al., 2016; Johnson & Zhang, 2013; Reddi et al., 2016; Lei et al., 2017). A well-known method, called Nesterov's accelerated gradient (NAG) originally designed for convex deterministic optimization (Nesterov, 2013; 1983; Nemirovskii et al., 1983), constructs the descent direction $\mathbf{\Delta}_t$ using the difference between the current iterate and the previous iterate. A recent work (Ghadimi & Lan, 2016) studied a generalization of NAG for non-convex stochastic programming. Similar in spirit to NAG, heavy-ball (HB) methods (Polyak, 1964; Ghadimi et al., 2015; Ochs et al., 2015; Yang et al., 2016) form the descent direction vector through a decaying sum of the previous gradient information. In addition to NAG and HB methods, stochastic variance reduced gradient (SVRG) methods integrate SGD with GD to acquire a hybrid descent direction of reduced variance (Johnson & Zhang, 2013; Reddi et al., 2016; Lei et al., 2017). Recently, certain accelerated version of perturbed gradient descent (PAGD) algorithm is also proposed in (Jin et al., 2017), which shows the fastest convergence rate among all Hessian free algorithms.

*Adaptive learning rate methods* accelerate ordinary SGD by using knowledge of the past gradients or second-order information into the current learning rate $\alpha_t$ (Becker et al., 1988; Duchi et al., 2011; Zeiler, 2012; Dauphin et al., 2015). In (Becker et al., 1988), the diagonal elements of the Hessian matrix were used to penalize a constant learning rate. However, acquiring the second-order information is computationally prohibitive. More recently, an adaptive subgradient method (i.e., AdaGrad) penalized the current gradient by dividing the square root of averaging of the squared gradient coordinates in earlier iterations (Duchi et al., 2011). Although AdaGrad works well when gradients are sparse, its convergence is only analyzed in the convex world. Other adaptive learning rate methods include Adadelta (Zeiler, 2012) and ESGD (Dauphin et al., 2015), which lacked theoretical investigation although some convergence improvement was shown in practice.

*Adaptive gradient methods* update the descent direction and the learning rate simultaneously using knowledge in the past, and thus enjoy dual advantages of momentum and adaptive learning rate methods. Algorithms of this family include RMSProp (Tieleman & Hinton, 2012), Nadam (Dozat, 2016), and Adam (Kingma & Ba, 2014). Among these, Adam has become the most widely-used method to train deep neural networks (DNNs). Specifically, Adam adopts *exponential* moving averages (with decaying/forgetting factors) of the past gradients to update the descent direction. It also uses inverse of exponential moving average of squared past gradients to adjust the learning rate. The work (Kingma & Ba, 2014) showed Adam converges with at most $O(1/\sqrt{T})$ rate for convex problems. However, the recent work (Reddi et al., 2018) pointed out the convergence issues of Adam even in the convex setting, and proposed a modified version of Adam (i.e., AMSGrad), which utilizes a non-increasing quadratic normalization and avoids the pitfalls of Adam. Although AMSGrad has made a significant progress toward understanding the theoretical behavior of adaptive gradient methods, the convergence analysis of (Reddi et al., 2018) only works for convex problems.

#### 6.1.1 Advantages and Disadvantages of Adaptive gradient method

In this section, we provide some additional experiments to demonstrate how specific Adam-type algorithms can perform better than SGD and how SGD can out perform Adam-type algorithms in different situations.

One possible benefit of adaptive gradient methods is the "sparse noise reduction" effect pointed out in Bernstein et al. (2018). Below we illustrate another possible practical advantage of adaptive gradient methods when applied to solve *non-convex problems*, which we refer to as *flexibility of stepsizes*.

To highlight ideas, let us take AMSGrad as an example, and compare it with SGD. First, in non-convex problems there can be multiple valleys with different curvatures. When using fixed stepsizes (or even a slowly diminishing stepsize), SGD can only converge to local optima in valleys with small curvature while AMSGrad and some other adaptive gradient algorithms can potentially converge to optima in valleys with relative high curvature (this may not be beneficial if one don't want to converge to a sharp local minimum). Second, the flexible choice of stepsizes implies less hyperparameter tuning and this coincides with the popular impression about original Adam.

We empirically demonstrate the flexible stepsizes property of AMSGrad using a deterministic quadratic problem. Consider a toy optimization problem $\min_x f(x)$, $f(x) = 100x^2$, the gradient is given by $200x$. For SGD (which reduces to gradient descent in this case) to converge, we must have $\alpha_t < 0.01$; for AMSGrad, $\hat{v}_t$ has a strong normalization effect and it allows the algorithm to use larger $\alpha_t$'s. We show the growth rate of different terms given in Theorem 3.1 for different stepsizes in Figure A1 to Figure A4 (where we choose $\beta_{1,t} = 0, \beta_{2,t} = 0.9$ for both Adam and AMSGrad). In Figure A1, $\alpha_t = 0.1$ and SGD diverges due to large $\alpha_t$, AMSGrad converges in this case, Adam is oscillating between two non-zero points. In Figure A2, stepsizes $\alpha_t$ is set to 0.01, SGD and Adam are oscillating, AMSGrad converges to 0. For Figure A3, SGD converges to 0 and AMSGrad is converging slower than SGD due to its smaller effective stepsizes, Adam is oscillating. One may wonder how diminishing stepsizes affects performance of the algorithms, this is shown in Figure A4 where $\alpha_t = 0.1/\sqrt{t}$, we can see SGD is diverging until stepsizes is small, AMSGrad is converging all the time, Adam appears to get stuck but it is actually converging very slowly due to diminishing stepsizes. This example shows AMSGrad can converge with a larger range of stepsizes compared with SGD.

From the figures, we can see that the term $\sum_{t=1}^{T} \|\alpha_t g_t / \sqrt{\hat{v}_t}\|^2$ is the key quantity that limits the convergence speed of algorithms in this case. In Figure A1, Figure A2, and early stage of Figure A4, the quantity is obviously a good sign of convergence speed. In Figure A3, since the difference of quantity between AMSGrad and SGD is compensated by the larger effective stepsizes of SGD and some problem independent constant, SGD converges faster. In fact, Figure A3 provides a case where AMSGrad does not perform well. Note that the normalization factor $\sqrt{\hat{v}_t}$ can be understood as imitating the largest Lipschitz constant along the way of optimization, so generally speaking dividing by this number makes the algorithm converge easier. However when the Lipschitz constant becomes smaller locally around a local optimal point, the stepsizes choice of AMSGrad dictates that $\sqrt{\hat{v}_t}$ does not change, resulting a small effective stepsizes. This could be mitigated by AdaGrad and its momentum variants which allows $\hat{v}_t$ to decrease when $g_t$ keeps decreasing.

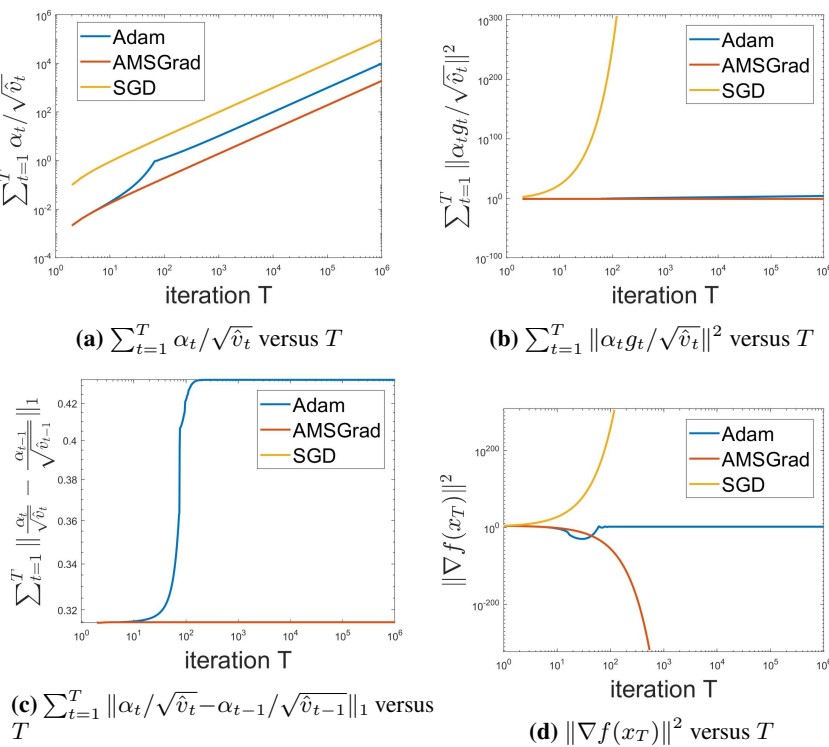

**(a)** $\sum_{t=1}^{T} \alpha_t / \sqrt{\hat{v}_t}$ versus $T$

**(b)** $\sum_{t=1}^{T} \|\alpha_t g_t / \sqrt{\hat{v}_t}\|^2$ versus $T$

**(c)** $\sum_{t=1}^{T} \|\alpha_t / \sqrt{\hat{v}_t} - \alpha_{t-1} / \sqrt{\hat{v}_{t-1}}\|_1$ versus $T$

**(d)** $\|\nabla f(x_T)\|^2$ versus $T$

**Figure A1:** Comparison of algorithms with $\alpha_t = 0.1$, we defined $\alpha_0 = 0$

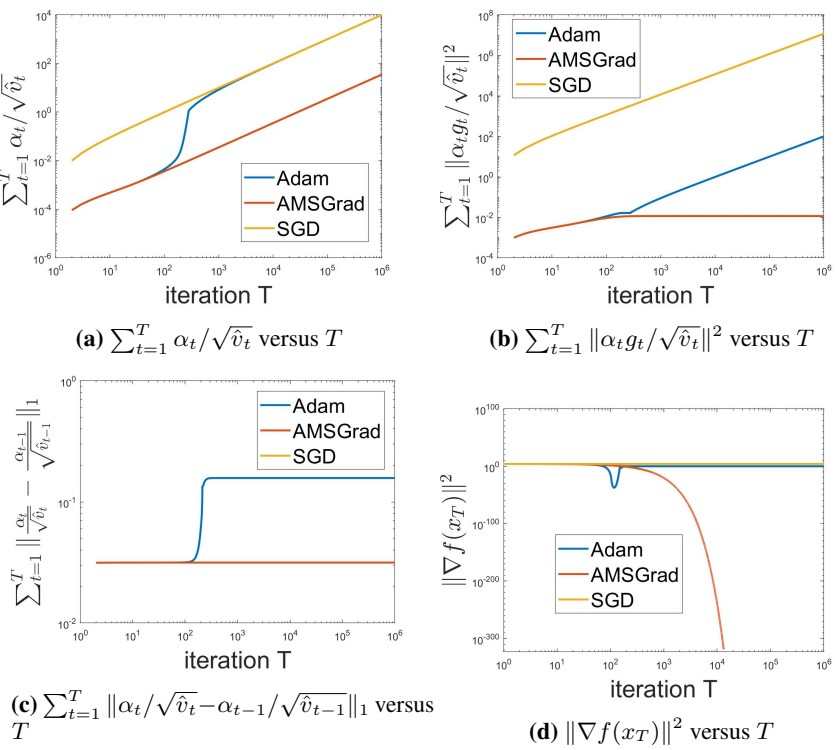

**(a)** $\sum_{t=1}^{T} \alpha_t / \sqrt{\hat{v}_t}$ versus $T$

**(b)** $\sum_{t=1}^{T} \|\alpha_t g_t / \sqrt{\hat{v}_t}\|^2$ versus $T$

**(c)** $\sum_{t=1}^{T} \|\alpha_t / \sqrt{\hat{v}_t} - \alpha_{t-1} / \sqrt{\hat{v}_{t-1}}\|_1$ versus $T$

**(d)** $\|\nabla f(x_T)\|^2$ versus $T$

**Figure A2:** Comparison of algorithms with $\alpha_t = 0.01$, we defined $\alpha_0 = 0$

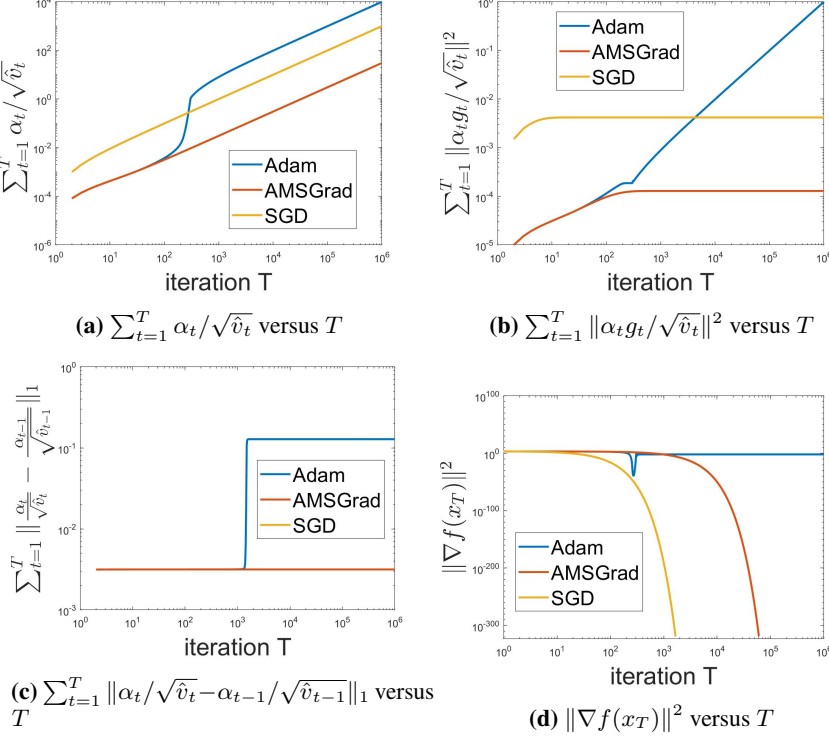

**(a)** $\sum_{t=1}^{T} \alpha_t / \sqrt{\hat{v}_t}$ versus $T$

**(b)** $\sum_{t=1}^{T} \|\alpha_t g_t / \sqrt{\hat{v}_t}\|^2$ versus $T$

**(c)** $\sum_{t=1}^{T} \|\alpha_t / \sqrt{\hat{v}_t} - \alpha_{t-1} / \sqrt{\hat{v}_{t-1}}\|_1$ versus $T$

**(d)** $\|\nabla f(x_T)\|^2$ versus $T$

**Figure A3:** Comparison of algorithms with $\alpha_t = 0.001$, we defined $\alpha_0 = 0$

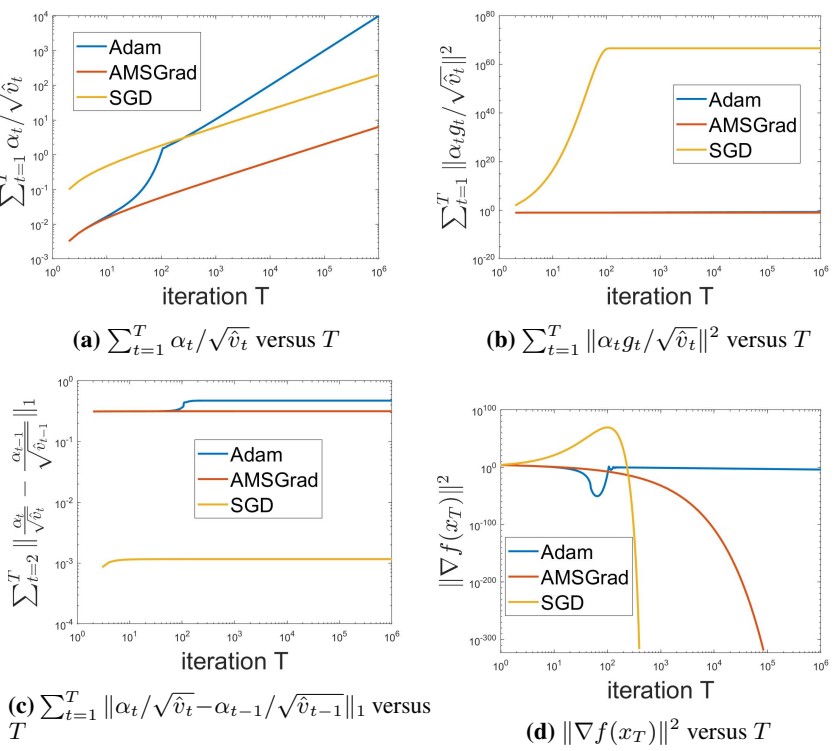

**Figure A4:** Comparison of algorithms with $\alpha_t = 0.1/\sqrt{t}$, we defined $\alpha_0 = 0$

### 6.1.2 DETAILS OF EXPERIMENTS ON MNIST AND CIFAR-10

In the experiment on MNIST, we consider a convolutional neural network (CNN), which includes 3 convolutional layers and 2 fully-connected layers. In convolutional layers, we adopt filters of sizes $6 \times 6 \times 1$ (with stride 1), $5 \times 5 \times 6$ (with stride 2), and $6 \times 6 \times 12$ (with stride 2), respectively. In both AMSGrad[2] and Adam, we set $\beta_1 = 0.9$ and $\beta_2 = 0.99$. In AdaFom, we set $\beta_1 = 0.9$. We choose 50 as the mini-batch size and the stepsize is choose to be $\alpha_t = 0.0001 + 0.003 e^{-t/2000}$.

The architecture of the CIFARNET that we are using is as below. The model starts with two convolutional layers with 32 and 64 kernels of size 3 x 3, followed by 2 x 2 max pooling and dropout with keep probability 0.25. The next layers are two convolutional layers with 128 kernels of size 3 x 3 and 2 x 2, respectively. Each of the two convolutional layers is followed by a 2 x 2 max pooling layer. The last layer is a fully connected layer with 1500 nodes. Dropout with keep probability 0.25 is added between the fully connected layer and the convolutional layer. All convolutional layers use ReLU activation and stride 1. The learning rate $\alpha_t$ of Adam and AMSGrad starts with 0.001 and decrease 10 times every 20 epochs. The learning rate of AdaGrad and AdaFom starts with 0.05 and decreases to 0.001 after 20 epochs and to 0.0001 after 40 epochs. These learning rates are tuned so that each algorithm has its best performance.

## 6.2 CONVERGENCE PROOF FOR GENERALIZED ADAM (ALGORITHM 1)

In this section, we present the convergence proof of Algorithm 1. We will first give several lemmas prior to proving Theorem 3.1.

---

[2] We customized our algorithms based on the open source code `https://github.com/taki0112/AMSGrad-Tensorflow`.

### 6.2.1 PROOF OF AUXILIARY LEMMAS

**Lemma 6.1.** *Let $x_0 \triangleq x_1$ in Algorithm 1, consider the sequence*

$$z_t = x_t + \frac{\beta_{1,t}}{1 - \beta_{1,t}}(x_t - x_{t-1}), \ \forall t \geq 1. \tag{9}$$

*Then the following holds true*

$$z_{t+1} - z_t = - \left( \frac{\beta_{1,t+1}}{1 - \beta_{1,t+1}} - \frac{\beta_{1,t}}{1 - \beta_{1,t}} \right) \alpha_t m_t / \sqrt{\hat{v}_t}$$

$$- \frac{\beta_{1,t}}{1 - \beta_{1,t}} \left( \frac{\alpha_t}{\sqrt{\hat{v}_t}} - \frac{\alpha_{t-1}}{\sqrt{\hat{v}_{t-1}}} \right) \odot m_{t-1} - \alpha_t g_t / \sqrt{\hat{v}_t}, \quad \forall\, t > 1$$

*and*

$$z_2 - z_1 = - \left( \frac{\beta_{1,2}}{1 - \beta_{1,2}} - \frac{\beta_{1,1}}{1 - \beta_{1,1}} \right) \alpha_1 m_1 / \sqrt{\hat{v}_1} - \alpha_1 g_1 / \sqrt{\hat{v}_1}.$$

**Proof.** [Proof of Lemma 6.1] By the update rules S1-S3 in Algorithm 1, we have when $t > 1$,

$$x_{t+1} - x_t = -\alpha_t m_t / \sqrt{\hat{v}_t}$$

$$\overset{S1}{=} - \alpha_t(\beta_{1,t} m_{t-1} + (1 - \beta_{1,t})g_t) / \sqrt{\hat{v}_t}$$

$$\overset{S3}{=} \beta_{1,t} \frac{\alpha_t}{\alpha_{t-1}} \frac{\sqrt{\hat{v}_{t-1}}}{\sqrt{\hat{v}_t}} \odot (x_t - x_{t-1}) - \alpha_t(1 - \beta_{1,t})g_t / \sqrt{\hat{v}_t}$$

$$= \beta_{1,t}(x_t - x_{t-1}) + \beta_{1,t} \left( \frac{\alpha_t}{\alpha_{t-1}} \frac{\sqrt{\hat{v}_{t-1}}}{\sqrt{\hat{v}_t}} - 1 \right) \odot (x_t - x_{t-1}) - \alpha_t(1 - \beta_{1,t})g_t / \sqrt{\hat{v}_t}$$

$$\overset{S3}{=} \beta_{1,t}(x_t - x_{t-1}) - \beta_{1,t} \left( \frac{\alpha_t}{\sqrt{\hat{v}_t}} - \frac{\alpha_{t-1}}{\sqrt{\hat{v}_{t-1}}} \right) \odot m_{t-1} - \alpha_t(1 - \beta_{1,t})g_t / \sqrt{\hat{v}_t}. \tag{10}$$

Since $x_{t+1} - x_t = (1 - \beta_{1,t})x_{t+1} + \beta_{1,t}(x_{t+1} - x_t) - (1 - \beta_{1,t})x_t$, based on (10) we have

$$(1 - \beta_{1,t})x_{t+1} + \beta_{1,t}(x_{t+1} - x_t)$$

$$= (1 - \beta_{1,t})x_t + \beta_{1,t}(x_t - x_{t-1}) - \beta_{1,t} \left( \frac{\alpha_t}{\sqrt{\hat{v}_t}} - \frac{\alpha_{t-1}}{\sqrt{\hat{v}_{t-1}}} \right) \odot m_{t-1} - \alpha_t(1 - \beta_{1,t})g_t / \sqrt{\hat{v}_t}.$$

Divide both sides by $1 - \beta_{1,t}$, we have

$$x_{t+1} + \frac{\beta_{1,t}}{1 - \beta_{1,t}}(x_{t+1} - x_t)$$

$$= x_t + \frac{\beta_{1,t}}{1 - \beta_{1,t}}(x_t - x_{t-1}) - \frac{\beta_{1,t}}{1 - \beta_{1,t}} \left( \frac{\alpha_t}{\sqrt{\hat{v}_t}} - \frac{\alpha_{t-1}}{\sqrt{\hat{v}_{t-1}}} \right) \odot m_{t-1} - \alpha_t g_t / \sqrt{\hat{v}_t}. \tag{11}$$

Define the sequence

$$z_t = x_t + \frac{\beta_{1,t}}{1 - \beta_{1,t}}(x_t - x_{t-1}).$$

Then (11) can be written as

$$z_{t+1} = z_t + \left( \frac{\beta_{1,t+1}}{1 - \beta_{1,t+1}} - \frac{\beta_{1,t}}{1 - \beta_{1,t}} \right) (x_{t+1} - x_t)$$

$$- \frac{\beta_{1,t}}{1 - \beta_{1,t}} \left( \frac{\alpha_t}{\sqrt{\hat{v}_t}} - \frac{\alpha_{t-1}}{\sqrt{\hat{v}_{t-1}}} \right) \odot m_{t-1} - \alpha_t g_t / \sqrt{\hat{v}_t}$$

$$= z_t - \left( \frac{\beta_{1,t+1}}{1 - \beta_{1,t+1}} - \frac{\beta_{1,t}}{1 - \beta_{1,t}} \right) \alpha_t m_t \sqrt{\hat{v}_t}$$

$$- \frac{\beta_{1,t}}{1 - \beta_{1,t}} \left( \frac{\alpha_t}{\sqrt{\hat{v}_t}} - \frac{\alpha_{t-1}}{\sqrt{\hat{v}_{t-1}}} \right) \odot m_{t-1} - \alpha_t g_t / \sqrt{\hat{v}_t}, \quad \forall t > 1,$$

where the second equality is due to $x_{t+1} - x_t = -\alpha_t m_t / \sqrt{\hat{v}_t}$.

For $t = 1$, we have $z_1 = x_1$ (due to $x_1 = x_0$), and

$$
\begin{aligned}
z_2 - z_1 =& x_2 + \frac{\beta_{1,2}}{1 - \beta_{1,2}}(x_2 - x_1) - x_1 \\
=& x_2 + \left( \frac{\beta_{1,2}}{1 - \beta_{1,2}} - \frac{\beta_{1,1}}{1 - \beta_{1,1}} \right)(x_2 - x_1) + \frac{\beta_{1,1}}{1 - \beta_{1,1}}(x_2 - x_1) - x_1 \\
=& \left( \frac{\beta_{1,2}}{1 - \beta_{1,2}} - \frac{\beta_{1,1}}{1 - \beta_{1,1}} \right)(-\alpha_1 m_1 / \sqrt{\hat{v}_1}) + \left( \frac{\beta_{1,1}}{1 - \beta_{1,1}} + 1 \right)(x_2 - x_1) \\
=& \left( \frac{\beta_{1,2}}{1 - \beta_{1,2}} - \frac{\beta_{1,1}}{1 - \beta_{1,1}} \right)(-\alpha_1 m_1 / \sqrt{\hat{v}_1}) + \frac{1}{1 - \beta_{1,1}}(-\alpha_1(1 - \beta_{1,1})g_1 / \sqrt{\hat{v}_1}) \\
=& -\left( \frac{\beta_{1,2}}{1 - \beta_{1,2}} - \frac{\beta_{1,1}}{1 - \beta_{1,1}} \right)(a_1 m_1 / \sqrt{\hat{v}_1}) - \alpha_1 g_1 / \sqrt{\hat{v}_1},
\end{aligned}
$$

where the forth equality holds due to (S1) and (S3) of Algorithm 1.

The proof is now complete. **Q.E.D.**

Without loss of generality, we initialize Algorithm 1 as below to simplify our analysis in what follows,

$$
\left( \frac{\alpha_1}{\sqrt{\hat{v}_1}} - \frac{\alpha_0}{\sqrt{\hat{v}_0}} \right) \odot m_0 = 0. \tag{12}
$$

**Lemma 6.2.** *Suppose that the conditions in Theorem 3.1 hold, then*

$$
E\left[ f(z_{t+1}) - f(z_1) \right] \le \sum_{i=1}^{6} T_i, \tag{13}
$$

*where*

$$
T_1 = -E\left[ \sum_{i=1}^{t} \langle \nabla f(z_i), \frac{\beta_{1,i}}{1 - \beta_{1,i}} \left( \frac{\alpha_i}{\sqrt{\hat{v}_i}} - \frac{\alpha_{i-1}}{\sqrt{\hat{v}_{i-1}}} \right) \odot m_{i-1} \rangle \right], \tag{14}
$$

$$
T_2 = -E\left[ \sum_{i=1}^{t} \alpha_i \langle \nabla f(z_i), g_i / \sqrt{\hat{v}_i} \rangle \right], \tag{15}
$$

$$
T_3 = -E\left[ \sum_{i=1}^{t} \langle \nabla f(z_i), \left( \frac{\beta_{1,i+1}}{1 - \beta_{1,i+1}} - \frac{\beta_{1,i}}{1 - \beta_{1,i}} \right) \alpha_i m_i / \sqrt{\hat{v}_i} \rangle \right], \tag{16}
$$

$$
T_4 = E\left[ \sum_{i=1}^{t} \frac{3}{2} L \left\| \left( \frac{\beta_{1,t+1}}{1 - \beta_{1,t+1}} - \frac{\beta_{1,t}}{1 - \beta_{1,t}} \right) \alpha_t m_t / \sqrt{v_t} \right\|^2 \right], \tag{17}
$$

$$
T_5 = E\left[ \sum_{i=1}^{t} \frac{3}{2} L \left\| \frac{\beta_{1,i}}{1 - \beta_{1,i}} \left( \frac{\alpha_t}{\sqrt{\hat{v}_i}} - \frac{\alpha_{i-1}}{\sqrt{\hat{v}_{i-1}}} \right) \odot m_{i-1} \right\|^2 \right], \tag{18}
$$

$$
T_6 = E\left[ \sum_{i=1}^{t} \frac{3}{2} L \left\| \alpha_i g_i / \sqrt{\hat{v}_i} \right\|^2 \right]. \tag{19}
$$

**Proof.** [Proof of Lemma 6.2] By the Lipschitz smoothness of $\nabla f$, we obtain

$$
f(z_{t+1}) \le f(z_t) + \langle \nabla f(z_t), d_t \rangle + \frac{L}{2} \| d_t \|^2, \tag{20}
$$

where $d_t = z_{t+1} - z_t$, and Lemma 6.1 together with (12) yield

$$
\begin{aligned}
d_t =& -\left( \frac{\beta_{1,t+1}}{1 - \beta_{1,t+1}} - \frac{\beta_{1,t}}{1 - \beta_{1,t}} \right) \alpha_t m_t / \sqrt{\hat{v}_t} \\
& - \frac{\beta_1}{1 - \beta_1} \left( \frac{\alpha_t}{\sqrt{\hat{v}_t}} - \frac{\alpha_{t-1}}{\sqrt{\hat{v}_{t-1}}} \right) \odot m_{t-1} - \alpha_t g_t / \sqrt{\hat{v}_t}, \quad \forall t \ge 1. \tag{21}
\end{aligned}
$$

Based on (20) and (21), we then have

$$
\begin{aligned}
E[f(z_{t+1}) - f(z_1)] = & E\left[\sum_{i=1}^{t} f(z_{i+1}) - f(z_i)\right] \\
\leq & E\left[\sum_{i=1}^{t} \langle \nabla f(z_i), d_i \rangle + \frac{L}{2}\|d_i\|^2\right] \\
= & - E\left[\sum_{i=1}^{t} \langle \nabla f(z_i), \frac{\beta_{1,i}}{1-\beta_{1,i}}\left(\frac{\alpha_i}{\sqrt{\hat{v}_i}} - \frac{\alpha_{i-1}}{\sqrt{\hat{v}_{i-1}}}\right) \odot m_{i-1}\rangle\right] \\
& - E\left[\sum_{i=1}^{t} \alpha_i \langle \nabla f(z_i), g_i/\sqrt{\hat{v}_i}\rangle\right] \\
& - E\left[\sum_{i=1}^{t} \langle \nabla f(z_i), \left(\frac{\beta_{1,i+1}}{1-\beta_{1,i+1}} - \frac{\beta_{1,i}}{1-\beta_{1,i}}\right)\alpha_i m_i/\sqrt{\hat{v}_i}\rangle\right] \\
& + E\left[\sum_{i=1}^{t} \frac{L}{2}\|d_i\|^2\right] = T_1 + T_2 + T_3 + + E\left[\sum_{i=1}^{t} \frac{L}{2}\|d_i\|^2\right], \qquad (22)
\end{aligned}
$$

where $\{T_i\}$ have been defined in (14)-(19). Further, using inequality $\|a+b+c\|^2 \leq 3\|a\|^2 + 3\|b\|^2 + 3\|c\|^2$ and (22), we have

$$
E\left[\sum_{i=1}^{t} \|d_i\|^2\right] \leq T_4 + T_5 + T_6.
$$

Substituting the above inequality into (22), we then obtain (13). $\qquad$ **Q.E.D.**

The next series of lemmas separately bound the terms on RHS of (13).

**Lemma 6.3.** *Suppose that the conditions in Theorem 3.1 hold, $T_1$ in (14) can be bounded as*

$$
\begin{aligned}
T_1 = & - E\left[\sum_{i=1}^{t} \langle \nabla f(z_i), \frac{\beta_{1,t}}{1-\beta_{1,t}}\left(\frac{\alpha_i}{\sqrt{\hat{v}_i}} - \frac{\alpha_{i-1}}{\sqrt{\hat{v}_{i-1}}}\right) \odot m_{i-1}\rangle\right] \\
\leq & H^2 \frac{\beta_1}{1-\beta_1} E\left[\sum_{i=2}^{t} \sum_{j=1}^{d} \left|\left(\frac{\alpha_i}{\sqrt{\hat{v}_i}} - \frac{\alpha_{i-1}}{\sqrt{\hat{v}_{i-1}}}\right)_j\right|\right]
\end{aligned}
$$

**Proof.** [Proof of Lemma 6.3] Since $\|g_t\| \leq H$, by the update rule of $m_t$, we have $\|m_t\| \leq H$, this can be proved by induction as below.

Recall that $m_t = \beta_{1,t} m_{t-1} + (1-\beta_{1,t})g_t$, suppose $\|m_{t-1}\| \leq H$, we have

$$
\|m_t\| \leq (\beta_{1,t} + (1-\beta_{1,t}))\max(\|g_t\|, \|m_{t-1}\|) = \max(\|g_t\|, \|m_{t-1}\|) \leq H, \qquad (23)
$$

then since $m_0 = 0$, we have $\|m_0\| \leq H$ which completes the induction.

Given $\|m_t\| \leq H$, we further have

$$
\begin{aligned}
T_1 = & - E\left[\sum_{i=2}^{t} \langle \nabla f(z_i), \frac{\beta_{1,t}}{1-\beta_{1,t}}\left(\frac{\alpha_i}{\sqrt{\hat{v}_i}} - \frac{\alpha_{i-1}}{\sqrt{\hat{v}_{i-1}}}\right) \odot m_{i-1}\rangle\right] \\
\leq & E\left[\sum_{i=1}^{t} \|\nabla f(z_i)\| \|m_{i-1}\| \left(\frac{1}{1-\beta_{1,t}} - 1\right)\sum_{j=1}^{d} \left|\left(\frac{\alpha_i}{\sqrt{\hat{v}_i}} - \frac{\alpha_{i-1}}{\sqrt{\hat{v}_{i-1}}}\right)_j\right|\right] \\
\leq & H^2 \frac{\beta_1}{1-\beta_1} E\left[\sum_{i=1}^{t} \sum_{j=1}^{d} \left|\left(\frac{\alpha_i}{\sqrt{\hat{v}_i}} - \frac{\alpha_{i-1}}{\sqrt{\hat{v}_{i-1}}}\right)_j\right|\right]
\end{aligned}
$$

where the first equality holds due to (12), and the last inequality is due to $\beta_1 \geq \beta_{1,i}$.

The proof is now complete. $\qquad$ **Q.E.D.**

**Lemma 6.4.** *Suppose the conditions in Theorem 3.1 hold. For $T_3$ in (16), we have*

$$T_3 = -E\left[\sum_{i=1}^t \langle \nabla f(z_i), \left(\frac{\beta_{1,i+1}}{1-\beta_{1,i+1}} - \frac{\beta_{1,i}}{1-\beta_{1,i}}\right)\alpha_i m_i/\sqrt{\hat{v}_i}\rangle\right]$$

$$\leq \left(\frac{\beta_1}{1-\beta_1} - \frac{\beta_{1,t+1}}{1-\beta_{1,t+1}}\right)\left(H^2 + G^2\right)$$

**Proof.** [Proof of Lemma 6.4]

$$T_3 \leq E\left[\sum_{i=1}^t \left|\frac{\beta_{1,i+1}}{1-\beta_{1,i+1}} - \frac{\beta_{1,i}}{1-\beta_{1,i}}\right| \frac{1}{2}\left(\|\nabla f(z_i)\|^2 + \|\alpha_i m_i/\sqrt{\hat{v}_i}\|^2\right)\right]$$

$$\leq E\left[\sum_{i=1}^t \left|\frac{\beta_{1,i+1}}{1-\beta_{1,i+1}} - \frac{\beta_{1,i}}{1-\beta_{1,i}}\right| \frac{1}{2}\left(H^2 + G^2\right)\right]$$

$$= \sum_{i=1}^t \left(\frac{\beta_{1,i}}{1-\beta_{1,i}} - \frac{\beta_{1,i+1}}{1-\beta_{1,i+1}}\right) \frac{1}{2}\left(H^2 + G^2\right)$$

$$\leq \left(\frac{\beta_1}{1-\beta_1} - \frac{\beta_{1,t+1}}{1-\beta_{1,t+1}}\right)\left(H^2 + G^2\right)$$

where the first inequality is due to $\langle a, b \rangle \leq \frac{1}{2}(\|a\|^2 + \|b\|^2)$, the second inequality is using due to upper bound on $\|\nabla f(x_t)\| \leq H$ and $\|\alpha_i m_i/\sqrt{\hat{v}_i}\| \leq G$ given by the assumptions in Theorem 3.1, the third equality is because $\beta_{1,t} \leq \beta_1$ and $\beta_{1,t}$ is non-increasing, the last inequality is due to telescope sum.

This completes the proof.                                                                                     **Q.E.D.**

**Lemma 6.5.** *Suppose the assumptions in Theorem 3.1 hold. For $T_4$ in (17), we have*

$$\frac{2}{3L}T_4 = E\left[\sum_{i=1}^t \left\|\left(\frac{\beta_{1,t+1}}{1-\beta_{1,t+1}} - \frac{\beta_{1,t}}{1-\beta_{1,t}}\right)\alpha_t m_t/\sqrt{v_t}\right\|^2\right]$$

$$\leq \left(\frac{\beta_1}{1-\beta_1} - \frac{\beta_{1,t+1}}{1-\beta_{1,t+1}}\right)^2 G^2$$

**Proof.** [Proof of Lemma 6.5] The proof is similar to the previous lemma.

$$\frac{2}{3L}T_4 = E\left[\sum_{i=1}^t \left(\frac{\beta_{1,t+1}}{1-\beta_{1,t+1}} - \frac{\beta_{1,t}}{1-\beta_{1,t}}\right)^2 \|\alpha_t m_t/\sqrt{v_t}\|^2\right]$$

$$\leq E\left[\sum_{i=1}^t \left(\frac{\beta_{1,t}}{1-\beta_{1,t}} - \frac{\beta_{1,t+1}}{1-\beta_{1,t+1}}\right)^2 G^2\right]$$

$$\leq \left(\frac{\beta_1}{1-\beta_1} - \frac{\beta_{1,t+1}}{1-\beta_{1,t+1}}\right)\sum_{i=1}^t\left(\frac{\beta_{1,t}}{1-\beta_{1,t}} - \frac{\beta_{1,t+1}}{1-\beta_{1,t+1}}\right) G^2$$

$$\leq \left(\frac{\beta_1}{1-\beta_1} - \frac{\beta_{1,t+1}}{1-\beta_{1,t+1}}\right)^2 G^2$$

where the first inequality is due to $\|\alpha_t m_t/\sqrt{v_t}\| \leq G$ by our assumptions, the second inequality is due to non-decreasing property of $\beta_{1,t}$ and $\beta_1 \geq \beta_{1,t}$, the last inequality is due to telescoping sum.

This completes the proof.                                                                                     **Q.E.D.**

**Lemma 6.6.** *Suppose the assumptions in Theorem 3.1 hold. For $T_5$ in (18), we have*

$$\frac{2}{3L}T_5 = E\left[\sum_{i=1}^t \left\|\frac{\beta_{1,i}}{1-\beta_{1,i}}\left(\frac{\alpha_t}{\sqrt{\hat{v}_i}} - \frac{\alpha_{i-1}}{\sqrt{\hat{v}_{i-1}}}\right)\odot m_{i-1}\right\|^2\right]$$

$$\leq \left(\frac{\beta_1}{1-\beta_1}\right)^2 H^2 E\left[\sum_{i=2}^t\sum_{j=1}^d\left(\frac{\alpha_t}{\sqrt{\hat{v}_i}} - \frac{\alpha_{i-1}}{\sqrt{\hat{v}_{i-1}}}\right)_j^2\right]$$

**Proof.** [Proof of Lemma 6.6]

$$\frac{2}{3L}T_5 \leq E\left[\sum_{i=2}^{t}\left(\frac{\beta_1}{1-\beta_1}\right)^2 \sum_{j=1}^{d}\left(\left(\frac{\alpha_t}{\sqrt{\hat{v}_i}} - \frac{\alpha_{i-1}}{\sqrt{\hat{v}_{i-1}}}\right)^2 (m_{i-1})_j^2\right)\right]$$

$$\leq \left(\frac{\beta_1}{1-\beta_1}\right)^2 H^2 E\left[\sum_{i=2}^{t}\sum_{j=1}^{d}\left(\frac{\alpha_t}{\sqrt{\hat{v}_i}} - \frac{\alpha_{i-1}}{\sqrt{\hat{v}_{i-1}}}\right)_j^2\right]$$

where the fist inequality is due to $\beta_1 \geq \beta_{1,t}$ and (12), the second inequality is due to $\|m_i\| < H$.

This completes the proof.                                                                           **Q.E.D.**

**Lemma 6.7.** *Suppose the assumptions in Theorem 3.1 hold. For $T_2$ in (15), we have*

$$T_2 = -E\left[\sum_{i=1}^{t}\alpha_i\langle\nabla f(z_i), g_i/\sqrt{\hat{v}_i}\rangle\right]$$

$$\leq \sum_{i=2}^{t}\frac{1}{2}\|\alpha_i g_i/\sqrt{\hat{v}_i}\|^2 + L^2\left(\frac{\beta_1}{1-\beta_1}\right)^2\left(\frac{1}{1-\beta_1}\right)^2 E\left[\sum_{i=1}^{t-1}\|\alpha_i g_i/\sqrt{\hat{v}_i}\|^2\right]$$

$$+ L^2 H^2\left(\frac{1}{1-\beta_1}\right)^2\left(\frac{\beta_1}{1-\beta_1}\right)^4 E\left[\sum_{j=1}^{d}\sum_{i=2}^{t-1}\left|\frac{\alpha_i}{\sqrt{\hat{v}_i}} - \frac{\alpha_{i-1}}{\sqrt{\hat{v}_{i-1}}}\right|_j^2\right]$$

$$+ 2H^2 E\left[\sum_{i=2}^{t}\sum_{j=1}^{d}\left|\left(\frac{\alpha_i}{\sqrt{\hat{v}_i}} - \frac{\alpha_{i-1}}{\sqrt{\hat{v}_{i-1}}}\right)_j\right|\right]$$

$$+ 2H^2 E\left[\sum_{j=1}^{d}(\alpha_1/\sqrt{\hat{v}_1})_j\right] - E\left[\sum_{i=1}^{t}\alpha_i\langle\nabla f(x_i), \nabla f(x_t)/\sqrt{\hat{v}_i}\rangle\right]. \tag{24}$$

**Proof.** [Proof of Lemma 6.7] Recall from the definition (9), we have

$$z_i - x_i = \frac{\beta_{1,i}}{1-\beta_{1,i}}(x_i - x_{i-1}) = -\frac{\beta_{1,i}}{1-\beta_{1,i}}\alpha_{i-1}m_{i-1}/\sqrt{\hat{v}_{i-1}} \tag{25}$$

Further we have $z_1 = x_1$ by definition of $z_1$. We have

$$T_2 = -E\left[\sum_{i=1}^{t}\alpha_i\langle\nabla f(z_i), g_i/\sqrt{\hat{v}_i}\rangle\right]$$

$$= -E\left[\sum_{i=1}^{t}\alpha_i\langle\nabla f(x_i), g_i/\sqrt{\hat{v}_i}\rangle\right] - E\left[\sum_{i=1}^{t}\alpha_i\langle\nabla f(z_i) - \nabla f(x_i), g_i/\sqrt{\hat{v}_i}\rangle\right]. \tag{26}$$

The second term of (26) can be bounded as

$$-E\left[\sum_{i=1}^{t}\alpha_i\langle\nabla f(z_i) - \nabla f(x_i), g_i/\sqrt{\hat{v}_i}\rangle\right]$$

$$\leq E\left[\sum_{i=2}^{t}\frac{1}{2}\|\nabla f(z_i) - \nabla f(x_i)\|^2 + \frac{1}{2}\|\alpha_i g_i/\sqrt{\hat{v}_i}\|^2\right]$$

$$\leq \frac{L^2}{2}T_7 + \frac{1}{2}E\left[\sum_{i=2}^{t}\|\alpha_i g_i/\sqrt{\hat{v}_i}\|^2\right], \tag{27}$$

where the first inequality is because $\langle a, b\rangle \leq \frac{1}{2}\left(\|a\|^2 + \|b\|^2\right)$ and the fact that $z_1 = x_1$, the second inequality is because

$$\|\nabla f(z_i) - \nabla f(x_i)\| \leq L\|z_i - x_i\| = L\|\frac{\beta_{1,t}}{1-\beta_{1,t}}\alpha_{i-1}m_{i-1}/\sqrt{\hat{v}_{i-1}}\|,$$

and $T_7$ is defined as

$$T_7 = E\left[\sum_{i=2}^{t}\left\|\frac{\beta_{1,i}}{1-\beta_{1,i}}\alpha_{i-1}m_{i-1}/\sqrt{\hat{v}_{i-1}}\right\|^2\right]. \tag{28}$$

We next bound the $T_7$ in (28), by update rule $m_i = \beta_{1,i}m_{i-1} + (1 - \beta_{1,i}g_i)$, we have $m_i = \sum_{k=1}^{i}[(\prod_{l=k+1}^{i}\beta_{1,l})(1-\beta_{1,k})g_k]$. Based on that, we obtain

$$T_7 \leq \left(\frac{\beta_1}{1-\beta_1}\right)^2 E\left[\sum_{i=2}^{t}\sum_{j=1}^{d}\left(\frac{\alpha_{i-1}m_{i-1}}{\sqrt{\hat{v}_{i-1}}}\right)_j^2\right]$$

$$= \left(\frac{\beta_1}{1-\beta_1}\right)^2 E\left[\sum_{i=2}^{t}\sum_{j=1}^{d}\left(\sum_{k=1}^{i-1}\frac{\alpha_{i-1}\left(\prod_{l=k+1}^{i-1}\beta_{1,l}\right)(1-\beta_{1,k})g_k}{\sqrt{\hat{v}_{i-1}}}\right)_j^2\right]$$

$$\leq 2\left(\frac{\beta_1}{1-\beta_1}\right)^2 \underbrace{E\left[\sum_{i=2}^{t}\sum_{j=1}^{d}\left(\sum_{k=1}^{i-1}\frac{\alpha_k\left(\prod_{l=k+1}^{i-1}\beta_{1,l}\right)(1-\beta_{1,k})g_k}{\sqrt{\hat{v}_k}}\right)_j^2\right]}_{T_8}$$

$$+ 2\left(\frac{\beta_1}{1-\beta_1}\right)^2 \underbrace{E\left[\sum_{i=2}^{t}\sum_{j=1}^{d}\left(\sum_{k=1}^{i-1}\left(\prod_{l=k+1}^{i-1}\beta_{1,l}\right)(1-\beta_{1,k})(g_k)_j\left(\frac{\alpha_{i-1}}{\sqrt{\hat{v}_{i-1}}}-\frac{\alpha_k}{\sqrt{\hat{v}_k}}\right)_j\right)^2\right]}_{T_9}$$

$$\tag{29}$$

where the first inequality is due to $\beta_{1,t} \leq \beta_1$, the second equality is by substituting expression of $m_t$, the last inequality is because $(a+b)^2 \leq 2(\|a\|^2 + \|b\|^2)$, and we have introduced $T_8$ and $T_9$ for ease of notation.

In (29), we first bound $T_8$ as below

$$T_8 = E\left[\sum_{i=2}^{t}\sum_{j=1}^{d}\sum_{k=1}^{i-1}\sum_{p=1}^{i-1}\left(\frac{\alpha_k g_k}{\sqrt{\hat{v}_k}}\right)_j\left(\prod_{l=k+1}^{i-1}\beta_{1,k}\right)(1-\beta_{1,k})\left(\frac{\alpha_p g_p}{\sqrt{\hat{v}_p}}\right)_j\left(\prod_{q=p+1}^{i-1}\beta_{1,p}\right)(1-\beta_{1,p})\right]$$

$$\overset{(i)}{\leq} E\left[\sum_{i=2}^{t}\sum_{j=1}^{d}\sum_{k=1}^{i-1}\sum_{p=1}^{i-1}\left(\beta_1^{i-1-k}\right)\left(\beta_1^{i-1-p}\right)\frac{1}{2}\left(\left(\frac{\alpha_k g_k}{\sqrt{\hat{v}_k}}\right)_j^2 + \left(\frac{\alpha_p g_p}{\sqrt{\hat{v}_p}}\right)_j^2\right)\right]$$

$$\overset{(ii)}{=} E\left[\sum_{i=2}^{t}\sum_{j=1}^{d}\sum_{k=1}^{i-1}\left(\beta_1^{i-1-k}\right)\left(\frac{\alpha_k g_k}{\sqrt{\hat{v}_k}}\right)_j^2\sum_{p=1}^{i-1}\left(\beta_1^{i-1-p}\right)\right]$$

$$\overset{(iii)}{\leq} \frac{1}{1-\beta_1}E\left[\sum_{i=2}^{t}\sum_{j=1}^{d}\sum_{k=1}^{i-1}\left(\beta_1^{i-1-k}\right)\left(\frac{\alpha_k g_k}{\sqrt{\hat{v}_k}}\right)_j^2\right]$$

$$\overset{(iv)}{=} \frac{1}{1-\beta_1}E\left[\sum_{k=1}^{t-1}\sum_{j=1}^{d}\sum_{i=k+1}^{t}\left(\beta_1^{i-1-k}\right)\left(\frac{\alpha_k g_k}{\sqrt{\hat{v}_k}}\right)_j^2\right]$$

$$\leq \left(\frac{1}{1-\beta_1}\right)^2 E\left[\sum_{k=1}^{t-1}\sum_{j=1}^{d}\left(\frac{\alpha_k g_k}{\sqrt{\hat{v}_k}}\right)_j^2\right] = \left(\frac{1}{1-\beta_1}\right)^2 E\left[\sum_{i=1}^{t-1}\|\alpha_i g_i/\sqrt{\hat{v}_i}\|^2\right] \tag{30}$$

where $(i)$ is due to $ab < \frac{1}{2}(a^2 + b^2)$ and follows from $\beta_{1,t} \leq \beta_1$ and $\beta_{1,t} \in [0,1)$, $(ii)$ is due to symmetry of $p$ and $k$ in the summation, $(iii)$ is because of $\sum_{p=1}^{i-1}\left(\beta_1^{i-1-p}\right) \leq \frac{1}{1-\beta_1}$, $(iv)$ is exchanging order of summation, and the second-last inequality is due to the similar reason as $(iii)$.

For the $T_9$ in (29), we have

$$
\begin{aligned}
T_9 =& E\left[\sum_{i=2}^{t}\sum_{j=1}^{d}\left(\sum_{k=1}^{i-1}\left(\prod_{l=k+1}^{i-1}\beta_{1,k}\right)(1-\beta_{1,k})(g_k)_j\left(\frac{\alpha_{i-1}}{\sqrt{\hat{v}_{i-1}}}-\frac{\alpha_k}{\sqrt{\hat{v}_k}}\right)_j\right)^2\right] \\
\leq& H^2 E\left[\sum_{i=2}^{t}\sum_{j=1}^{d}\left(\sum_{k=1}^{i-1}\left(\prod_{l=k+1}^{i-1}\beta_{1,k}\right)\left|\frac{\alpha_{i-1}}{\sqrt{\hat{v}_{i-1}}}-\frac{\alpha_k}{\sqrt{\hat{v}_k}}\right|_j\right)^2\right] \\
\leq& H^2 E\left[\sum_{i=1}^{t-1}\sum_{j=1}^{d}\left(\sum_{k=1}^{i}\beta_1^{i-k}\left|\frac{\alpha_i}{\sqrt{\hat{v}_i}}-\frac{\alpha_k}{\sqrt{\hat{v}_k}}\right|_j\right)^2\right] \\
\leq& H^2 E\left[\sum_{i=1}^{t-1}\sum_{j=1}^{d}\left(\sum_{k=1}^{i}\beta_1^{i-k}\sum_{l=k+1}^{i}\left|\frac{\alpha_l}{\sqrt{\hat{v}_l}}-\frac{\alpha_{l-1}}{\sqrt{\hat{v}_{l-1}}}\right|_j\right)^2\right]
\end{aligned}
\tag{31}
$$

where the first inequality holds due to $\beta_{1,k} < 1$ and $|(g_k)_j| \leq H$, the second inequality holds due to $\beta_{1,k} \leq \beta_1$, and the last inequality applied the triangle inequality. For RHS of (31), using Lemma 6.8 (that will be proved later) with $a_i = \left|\frac{\alpha_i}{\sqrt{\hat{v}_i}}-\frac{\alpha_{i-1}}{\sqrt{\hat{v}_{i-1}}}\right|_j$, we further have

$$
\begin{aligned}
T_9 \leq& H^2 E\left[\sum_{i=1}^{t-1}\sum_{j=1}^{d}\left(\sum_{k=1}^{i}\beta_1^{i-k}\sum_{l=k+1}^{i}\left|\frac{\alpha_l}{\sqrt{\hat{v}_l}}-\frac{\alpha_{l-1}}{\sqrt{\hat{v}_{l-1}}}\right|_j\right)^2\right] \\
\leq& H^2\left(\frac{1}{1-\beta_1}\right)^2\left(\frac{\beta_1}{1-\beta_1}\right)^2 E\left[\sum_{j=1}^{d}\sum_{i=2}^{t-1}\left|\frac{\alpha_l}{\sqrt{\hat{v}_l}}-\frac{\alpha_{l-1}}{\sqrt{\hat{v}_{l-1}}}\right|_j^2\right]
\end{aligned}
\tag{32}
$$

Based on (27), (29), (30) and (32), we can then bound the second term of (26) as

$$
\begin{aligned}
& -E\left[\sum_{i=1}^{t}\alpha_i\langle\nabla f(z_i)-\nabla f(x_i), g_i/\sqrt{\hat{v}_i}\rangle\right] \\
\leq& L^2\left(\frac{\beta_1}{1-\beta_1}\right)^2\left(\frac{1}{1-\beta_1}\right)^2 E\left[\sum_{i=1}^{t-1}\|\alpha_i g_i/\sqrt{\hat{v}_i}\|^2\right] \\
& + L^2 H^2\left(\frac{1}{1-\beta_1}\right)^2\left(\frac{\beta_1}{1-\beta_1}\right)^4 E\left[\sum_{j=1}^{d}\sum_{i=2}^{t-1}\left|\frac{\alpha_l}{\sqrt{\hat{v}_l}}-\frac{\alpha_{l-1}}{\sqrt{\hat{v}_{l-1}}}\right|_j^2\right] \\
& + \frac{1}{2}E\left[\sum_{i=2}^{t}\|\alpha_i g_i/\sqrt{\hat{v}_i}\|^2\right].
\end{aligned}
\tag{33}
$$

Let us turn to the first term in (26). Reparameterize $g_t$ as $g_t = \nabla f(x_t) + \delta_t$ with $E[\delta_t] = 0$, we have

$$
\begin{aligned}
& E\left[\sum_{i=1}^{t}\alpha_i\langle\nabla f(x_i), g_i/\sqrt{\hat{v}_i}\rangle\right] \\
=& E\left[\sum_{i=1}^{t}\alpha_i\langle\nabla f(x_i), (\nabla f(x_i)+\delta_i)/\sqrt{\hat{v}_i}\rangle\right] \\
=& E\left[\sum_{i=1}^{t}\alpha_i\langle\nabla f(x_i), \nabla f(x_i)/\sqrt{\hat{v}_i}\rangle\right] + E\left[\sum_{i=1}^{t}\alpha_i\langle\nabla f(x_i), \delta_i/\sqrt{\hat{v}_i}\rangle\right].
\end{aligned}
\tag{34}
$$

It can be seen that the first term in RHS of (34) is the desired descent quantity, the second term is a bias term to be bounded. For the second term in RHS of (34), we have

$$
E\left[\sum_{i=1}^{t} \alpha_i \langle \nabla f(x_i), \delta_i/\sqrt{\hat{v}_i}\rangle\right]
$$

$$
=E\left[\sum_{i=2}^{t}\langle \nabla f(x_i), \delta_i \odot (\alpha_i/\sqrt{\hat{v}_i} - \alpha_{i-1}/\sqrt{\hat{v}_{i-1}})\rangle\right] + E\left[\sum_{i=2}^{t}\alpha_{i-1}\langle \nabla f(x_i), \delta_i \odot (1/\sqrt{\hat{v}_{i-1}})\rangle\right]
$$

$$
+ E\left[\alpha_1 \langle \nabla f(x_1), \delta_1/\sqrt{\hat{v}_1}\rangle\right]
$$

$$
\geq E\left[\sum_{i=2}^{t}\langle \nabla f(x_i), \delta_i \odot (\alpha_i/\sqrt{\hat{v}_i} - \alpha_{i-1}/\sqrt{\hat{v}_{i-1}})\rangle\right] - 2H^2 E\left[\sum_{j=1}^{d}(\alpha_1/\sqrt{\hat{v}_1})_j\right] \tag{35}
$$

where the last equation is because given $x_i, \hat{v}_{i-1}$, $E\left[\delta_i \odot (1/\sqrt{\hat{v}_{i-1}})|x_i, \hat{v}_{i-1}\right] = 0$ and $\|\delta_i\| \leq 2H$ due to $\|g_i\| \leq H$ and $\|\nabla f(x_i)\| \leq H$ based on Assumptions A2 and A3. Further, we have

$$
E\left[\sum_{i=2}^{t}\langle \nabla f(x_i), \delta_t \odot (\alpha_i/\sqrt{\hat{v}_i} - \alpha_{i-1}/\sqrt{\hat{v}_{i-1}})\rangle\right]
$$

$$
=E\left[\sum_{i=2}^{t}\sum_{j=1}^{d}(\nabla f(x_i))_j(\delta_t)_j(\alpha_i/(\sqrt{\hat{v}_i})_j - \alpha_{i-1}/(\sqrt{\hat{v}_{i-1}})_j)\right]
$$

$$
\geq -E\left[\sum_{i=2}^{t}\sum_{j=1}^{d}|(\nabla f(x_i))_j|\,|(\delta_t)_j|\left|(\alpha_i/(\sqrt{\hat{v}_i})_j - \alpha_{i-1}/(\sqrt{\hat{v}_{i-1}})_j)\right|\right]
$$

$$
\geq -2H^2 E\left[\sum_{i=2}^{t}\sum_{j=1}^{d}\left|(\alpha_i/(\sqrt{\hat{v}_i})_j - \alpha_{i-1}/(\sqrt{\hat{v}_{i-1}})_j)\right|\right] \tag{36}
$$

Substituting (35) and (36) into (34), we then bound the first term of (26) as

$$
-E\left[\sum_{i=1}^{t}\alpha_i \langle \nabla f(x_i), g_i/\sqrt{\hat{v}_i}\rangle\right]
$$

$$
\leq 2H^2 E\left[\sum_{i=2}^{t}\sum_{j=1}^{d}\left|(\alpha_i/(\sqrt{\hat{v}_i})_j - \alpha_{i-1}/(\sqrt{\hat{v}_{i-1}})_j)\right|\right] + 2H^2 E\left[\sum_{j=1}^{d}(\alpha_1/\sqrt{\hat{v}_1})_j\right]
$$

$$
-E\left[\sum_{i=1}^{t}\alpha_i \langle \nabla f(x_i), \nabla f(x_i)/\sqrt{\hat{v}_i}\rangle\right] \tag{37}
$$

We finally apply (37) and (33) to obtain (24). The proof is now complete. **Q.E.D**.

**Lemma 6.8.** *For $a_i \geq 0$, $\beta \in [0, 1)$, and $b_i = \sum_{k=1}^{i}\beta^{i-k}\sum_{l=k+1}^{i} a_l$, we have*

$$
\sum_{i=1}^{t} b_i^2 \leq \left(\frac{1}{1-\beta}\right)^2 \left(\frac{\beta}{1-\beta}\right)^2 \sum_{i=2}^{t} a_i^2
$$

**Proof.** [Proof of Lemma 6.8] The result is proved by following

$$
\sum_{i=1}^{t} b_i^2 = \sum_{i=1}^{t} \left( \sum_{k=1}^{i} \beta^{i-k} \sum_{l=k+1}^{i} a_l \right)^2
$$

$$
\overset{(i)}{=} \sum_{i=1}^{t} \left( \sum_{l=2}^{i} \sum_{k=1}^{l-1} \beta^{i-k} a_l \right)^2 = \sum_{i=1}^{t} \left( \sum_{l=2}^{i} \beta^{i-l+1} a_l \sum_{k=1}^{l-1} \beta^{l-1-k} \right)^2
$$

$$
\overset{(ii)}{\leq} \left( \frac{1}{1-\beta} \right)^2 \sum_{i=1}^{t} \left( \sum_{l=2}^{i} \beta^{i-l+1} a_l \right)^2 = \left( \frac{1}{1-\beta} \right)^2 \sum_{i=1}^{t} \left( \sum_{l=2}^{i} \sum_{m=2}^{i} \beta^{i-l+1} a_l \beta^{i-m+1} a_m \right)
$$

$$
\overset{(iii)}{\leq} \left( \frac{1}{1-\beta} \right)^2 \sum_{i=1}^{t} \sum_{l=2}^{i} \sum_{m=2}^{i} \beta^{i-l+1} \beta^{i-m+1} \frac{1}{2} \left( a_l^2 + a_m^2 \right)
$$

$$
\overset{(iv)}{=} \left( \frac{1}{1-\beta} \right)^2 \sum_{i=1}^{t} \sum_{l=2}^{i} \sum_{m=2}^{i} \beta^{i-l+1} \beta^{i-m+1} a_l^2 \overset{(v)}{\leq} \left( \frac{1}{1-\beta} \right)^2 \frac{\beta}{1-\beta} \sum_{l=2}^{t} \sum_{i=l}^{t} \beta^{i-l+1} a_l^2
$$

$$
\leq \left( \frac{1}{1-\beta} \right)^2 \left( \frac{\beta}{1-\beta} \right)^2 \sum_{l=2}^{t} a_l^2
$$

where $(i)$ is by changing order of summation, $(ii)$ is due to $\sum_{k=1}^{l-1} \beta^{l-1-k} \leq \frac{1}{1-\beta}$, $(iii)$ is by the fact that $ab \leq \frac{1}{2}(a^2 + b^2)$, $(iv)$ is due to symmetry of $a_l$ and $a_m$ in the summation, $(v)$ is because $\sum_{m=2}^{i} \beta^{i-m+1} \leq \frac{\beta}{1-\beta}$ and the last inequality is for similar reason.

This completes the proof. **Q.E.D.**

### 6.2.2   PROOF OF THEOREM 3.1

**Proof.** [Proof of Theorem 3.1] We combine Lemma 6.2, Lemma 6.3, Lemma 6.4, Lemma 6.5, Lemma 6.6, and Lemma 6.7 to bound the overall expected descent of the objective. First, from Lemma 6.2, we have

$$
E\left[f(z_{t+1}) - f(z_1)\right] \leq \sum_{i=1}^{6} T_i
$$

$$
= - E\left[ \sum_{i=1}^{t} \langle \nabla f(z_i), \frac{\beta_{1,i}}{1-\beta_{1,i}} \left( \frac{\alpha_i}{\sqrt{\hat{v}_i}} - \frac{\alpha_{i-1}}{\sqrt{\hat{v}_{i-1}}} \right) \odot m_{i-1} \rangle \right] - E\left[ \sum_{i=1}^{t} \alpha_i \langle \nabla f(z_i), g_i/\sqrt{\hat{v}_i} \rangle \right]
$$

$$
- E\left[ \sum_{i=1}^{t} \langle \nabla f(z_i), \left( \frac{\beta_{1,i+1}}{1-\beta_{1,i+1}} - \frac{\beta_{1,i}}{1-\beta_{1,i}} \right) \alpha_i m_i/\sqrt{\hat{v}_i} \rangle \right]
$$

$$
+ E\left[ \sum_{i=1}^{t} \frac{3}{2} L \left\| \left( \frac{\beta_{1,t+1}}{1-\beta_{1,t+1}} - \frac{\beta_{1,t}}{1-\beta_{1,t}} \right) \alpha_t m_t/\sqrt{\hat{v}_t} \right\|^2 \right]
$$

$$
+ E\left[ \sum_{i=1}^{t} \frac{3}{2} L \left\| \frac{\beta_{1,i}}{1-\beta_{1,i}} \left( \frac{\alpha_t}{\sqrt{\hat{v}_i}} - \frac{\alpha_{i-1}}{\sqrt{\hat{v}_{i-1}}} \right) \odot m_{i-1} \right\|^2 \right] + E\left[ \sum_{i=1}^{t} \frac{3}{2} L \left\| \alpha_i g_i/\sqrt{\hat{v}_i} \right\|^2 \right]
$$

$$
\tag{38}
$$

Then from above inequality and Lemma 6.3, Lemma 6.4, Lemma 6.5, Lemma 6.6, Lemma 6.7, we get

$$
\begin{aligned}
& E\left[f(z_{t+1}) - f(z_1)\right] \\
\leq & H^2 \frac{\beta_1}{1-\beta_1} E\left[\sum_{i=2}^{t}\sum_{j=1}^{d}\left|\left(\frac{\alpha_i}{\sqrt{\hat{v}_i}} - \frac{\alpha_{i-1}}{\sqrt{\hat{v}_{i-1}}}\right)_j\right|\right] \\
& + \left(\frac{\beta_1}{1-\beta_1} - \frac{\beta_{1,t+1}}{1-\beta_{1,t+1}}\right)\left(H^2+G^2\right) + \left(\frac{\beta_1}{1-\beta_1} - \frac{\beta_{1,t+1}}{1-\beta_{1,t+1}}\right)^2 G^2 \\
& + \left(\frac{\beta_1}{1-\beta_1}\right)^2 H^2 E\left[\sum_{i=2}^{t}\sum_{j=1}^{d}\left(\frac{\alpha_i}{\sqrt{\hat{v}_i}} - \frac{\alpha_{i-1}}{\sqrt{\hat{v}_{i-1}}}\right)_j^2\right] + E\left[\sum_{i=1}^{t}\frac{3}{2}L\left\|\alpha_i g_i/\sqrt{\hat{v}_i}\right\|^2\right] \\
& + E\left[\sum_{i=2}^{t}\frac{1}{2}\|\alpha_i g_i/\sqrt{\hat{v}_i}\|^2\right] + L^2\frac{\beta_1}{1-\beta_1}\left(\frac{1}{1-\beta_1}\right)^2 E\left[\sum_{k=1}^{t-1}\sum_{j=1}^{d}\left(\frac{\alpha_k g_k}{\sqrt{\hat{v}_k}}\right)_j^2\right] \\
& + L^2 H^2\left(\frac{1}{1-\beta_1}\right)^2\left(\frac{\beta_1}{1-\beta_1}\right)^4 E\left[\sum_{j=1}^{d}\sum_{i=2}^{t-1}\left(\frac{\alpha_i}{\sqrt{\hat{v}_i}} - \frac{\alpha_{i-1}}{\sqrt{\hat{v}_{i-1}}}\right)_j^2\right] + 2H^2 E\left[\sum_{i=2}^{t}\sum_{j=1}^{d}\left|\left(\frac{\alpha_i}{\sqrt{\hat{v}_i}} - \frac{\alpha_{i-1}}{\sqrt{\hat{v}_{i-1}}}\right)_j\right|\right] \\
& + 2H^2 E\left[\sum_{j=1}^{d}(\alpha_1/\sqrt{\hat{v}_1})_j\right] - E\left[\sum_{i=1}^{t}\alpha_i\langle\nabla f(x_i), \nabla f(x_i)/\sqrt{\hat{v}_i}\rangle\right]
\end{aligned}
$$

By merging similar terms in above inequality, we further have

$$
\begin{aligned}
& E\left[f(z_{t+1}) - f(z_1)\right] \\
\leq & \left(H^2\frac{\beta_1}{1-\beta_1} + 2H^2\right) E\left[\sum_{i=2}^{t}\sum_{j=1}^{d}\left|\left(\frac{\alpha_i}{\sqrt{\hat{v}_i}} - \frac{\alpha_{i-1}}{\sqrt{\hat{v}_{i-1}}}\right)_j\right|\right] \\
& + \left(1 + L^2\left(\frac{1}{1-\beta_1}\right)^2\left(\frac{\beta_1}{1-\beta_1}\right)^2\right) H^2\left(\frac{\beta_1}{1-\beta_1}\right)^2 E\left[\sum_{j=1}^{d}\sum_{i=2}^{t-1}\left(\frac{\alpha_i}{\sqrt{\hat{v}_i}} - \frac{\alpha_{i-1}}{\sqrt{\hat{v}_{i-1}}}\right)_j^2\right] \\
& + \left(\frac{3}{2}L + \frac{1}{2} + L^2\frac{\beta_1}{1-\beta_1}\left(\frac{1}{1-\beta_1}\right)^2\right) E\left[\sum_{i=1}^{t}\left\|\alpha_i g_i/\sqrt{\hat{v}_i}\right\|^2\right] \\
& + \left(\frac{\beta_1}{1-\beta_1} - \frac{\beta_{1,t+1}}{1-\beta_{1,t+1}}\right)\left(H^2+G^2\right) + \left(\frac{\beta_1}{1-\beta_1} - \frac{\beta_{1,t+1}}{1-\beta_{1,t+1}}\right)^2 G^2 \\
& + 2H^2 E\left[\sum_{j=1}^{d}(\alpha_1/\sqrt{\hat{v}_1})_j\right] - E\left[\sum_{i=1}^{t}\alpha_i\langle\nabla f(x_i), \nabla f(x_i)/\sqrt{\hat{v}_i}\rangle\right]
\end{aligned}
\tag{39}
$$

Rearranging (39), we have

$$E\left[\sum_{i=1}^{t}\alpha_i\langle\nabla f(x_i),\nabla f(x_i)/\sqrt{\hat{v}_i}\rangle\right]$$

$$\leq\left(H^2\frac{\beta_1}{1-\beta_1}+2H^2\right)E\left[\sum_{i=2}^{t}\sum_{j=1}^{d}\left|\left(\frac{\alpha_i}{\sqrt{\hat{v}_i}}-\frac{\alpha_{i-1}}{\sqrt{\hat{v}_{i-1}}}\right)_j\right|\right]+$$

$$+\left(1+L^2\left(\frac{1}{1-\beta_1}\right)^2\left(\frac{\beta_1}{1-\beta_1}\right)\right)H^2\left(\frac{\beta_1}{1-\beta_1}\right)^2E\left[\sum_{j=1}^{d}\sum_{i=2}^{t-1}\left(\frac{\alpha_i}{\sqrt{\hat{v}_i}}-\frac{\alpha_{i-1}}{\sqrt{\hat{v}_{i-1}}}\right)_j^2\right]$$

$$+\left(\frac{3}{2}L+\frac{1}{2}+L^2\frac{\beta_1}{1-\beta_1}\left(\frac{1}{1-\beta_1}\right)^2\right)E\left[\sum_{i=1}^{t}\left\|\alpha_i g_i/\sqrt{\hat{v}_i}\right\|^2\right]$$

$$+\left(\frac{\beta_1}{1-\beta_1}-\frac{\beta_{1,t+1}}{1-\beta_{1,t+1}}\right)(H^2+G^2)+\left(\frac{\beta_1}{1-\beta_1}-\frac{\beta_{1,t+1}}{1-\beta_{1,t+1}}\right)^2G^2$$

$$+2H^2E\left[\sum_{j=1}^{d}(\alpha_1/\sqrt{\hat{v}_1})_j\right]+E\left[f(z_1)-f(z_{t+1})\right]$$

$$\leq E\left[C_1\sum_{i=1}^{t}\left\|\alpha_t g_t/\sqrt{\hat{v}_t}\right\|^2+C_2\sum_{i=2}^{t}\left\|\frac{\alpha_i}{\sqrt{\hat{v}_i}}-\frac{\alpha_{i-1}}{\sqrt{\hat{v}_{i-1}}}\right\|_1+C_3\sum_{i=2}^{t-1}\left\|\frac{\alpha_i}{\sqrt{\hat{v}_i}}-\frac{\alpha_{i-1}}{\sqrt{\hat{v}_{i-1}}}\right\|^2\right]+C_4$$

where

$$C_1\triangleq\left(\frac{3}{2}L+\frac{1}{2}+L^2\frac{\beta_1}{1-\beta_1}\left(\frac{1}{1-\beta_1}\right)^2\right)$$

$$C_2\triangleq\left(H^2\frac{\beta_1}{1-\beta_1}+2H^2\right)$$

$$C_3\triangleq\left(1+L^2\left(\frac{1}{1-\beta_1}\right)^2\left(\frac{\beta_1}{1-\beta_1}\right)\right)H^2\left(\frac{\beta_1}{1-\beta_1}\right)^2$$

$$C_4\triangleq\left(\frac{\beta_1}{1-\beta_1}\right)(H^2+G^2)+\left(\frac{\beta_1}{1-\beta_1}\right)^2G^2$$
$$+2H^2E\left[\|\alpha_1/\sqrt{\hat{v}_1}\|_1\right]+E\left[f(z_1)-f(z^*)\right]$$

and $z^*$ is an optimal of $f$, i.e. $z^*\in\arg\min_z f(z)$.

Using the fact that $(\alpha_i/\sqrt{\hat{v}_i})_j\geq\gamma_i,\forall j$ by definition, inequality (4) directly follows.

This completes the proof.                                   Q.E.D.

### 6.2.3   PROOF OF COROLLARY 3.1

**Proof.** [Proof of Corollary 3.1]

---

**Algorithm 3. AMSGrad**

(S0). Define $m_0=0$, $v_0=0$, $\hat{v}_0=0$;
For $t=1,\cdots,T$, do
   (S1). $m_t=\beta_{1,t}m_{t-1}+(1-\beta_{1,t})g_t$
   (S2). $v_t=\beta_2 v_{t-1}+(1-\beta_2)g_t^2$
   (S3). $\hat{v}_t=\max\{\hat{v}_{t-1},v_t\}$
   (S4). $x_{t+1}=x_t-\alpha_t m_t/\sqrt{\hat{v}_t}$
End

---

We first bound non-constant terms in RHS of (3), which is given by

$$E\left[C_1\sum_{t=1}^{T}\left\|\alpha_t g_t/\sqrt{\hat{v}_t}\right\|^2 + C_2\sum_{t=2}^{T}\left\|\frac{\alpha_t}{\sqrt{\hat{v}_t}} - \frac{\alpha_{t-1}}{\sqrt{\hat{v}_{t-1}}}\right\|_1 + C_3\sum_{t=2}^{T-1}\left\|\frac{\alpha_t}{\sqrt{\hat{v}_t}} - \frac{\alpha_{t-1}}{\sqrt{\hat{v}_{t-1}}}\right\|^2\right] + C_4.$$

For the term with $C_1$, assume $\min_{j\in[d]}(\sqrt{\hat{v}_1})_j \geq c > 0$ (this is natural since if it is 0, division by 0 error will happen), we have

$$E\left[\sum_{t=1}^{T}\left\|\alpha_t g_t/\sqrt{\hat{v}_t}\right\|^2\right]$$

$$\leq E\left[\sum_{t=1}^{T}\|\alpha_t g_t/c\|^2\right] = E\left[\sum_{t=1}^{T}\left\|\frac{1}{\sqrt{t}}g_t/c\right\|^2\right] = E\left[\sum_{t=1}^{T}\left(\frac{1}{c\sqrt{t}}\right)^2\|g_t\|^2\right]$$

$$\leq H^2/c^2\sum_{t=1}^{T}\frac{1}{t} \leq H^2/c^2(1+\log T)$$

where the first inequality is due to $(\hat{v}_t)_j \geq (\hat{v}_{t-1})_j$, and the last inequality is due to $\sum_{t=1}^{T}1/t \leq 1+\log T$.

For the term with $C_2$, we have

$$E\left[\sum_{t=2}^{T}\left\|\frac{\alpha_t}{\sqrt{\hat{v}_t}} - \frac{\alpha_{t-1}}{\sqrt{\hat{v}_{t-1}}}\right\|_1\right] = E\left[\sum_{j=1}^{d}\sum_{t=2}^{T}\left(\frac{\alpha_{t-1}}{(\sqrt{\hat{v}_{t-1}})_j} - \frac{\alpha_t}{(\sqrt{\hat{v}_t})_j}\right)\right]$$

$$= E\left[\sum_{j=1}^{d}\left(\frac{\alpha_1}{(\sqrt{\hat{v}_1})_j} - \frac{\alpha_T}{(\sqrt{\hat{v}_T})_j}\right)\right] \leq E\left[\sum_{j=1}^{d}\frac{\alpha_1}{(\sqrt{\hat{v}_1})_j}\right] \leq d/c \quad (40)$$

where the first equality is due to $(\hat{v}_t)_j \geq (\hat{v}_{t-1})_j$ and $\alpha_t \leq \alpha_{t-1}$, and the second equality is due to telescope sum.

For the term with $C_3$, we have

$$E\left[\sum_{t=2}^{T-1}\left\|\frac{\alpha_t}{\sqrt{\hat{v}_t}} - \frac{\alpha_{t-1}}{\sqrt{\hat{v}_{t-1}}}\right\|^2\right]$$

$$\leq E\left[\frac{1}{c}\sum_{t=2}^{T-1}\left\|\frac{\alpha_t}{\sqrt{\hat{v}_t}} - \frac{\alpha_{t-1}}{\sqrt{\hat{v}_{t-1}}}\right\|_1\right]$$

$$\leq d/c^2$$

where the first inequality is due to $|(\alpha_t/\sqrt{\hat{v}_t} - \alpha_{t-1}/\sqrt{\hat{v}_{t-1}})_j| \leq 1/c$.

Then we have for AMSGRAD,

$$E\left[C_1\sum_{t=1}^{T}\left\|\alpha_t g_t/\sqrt{\hat{v}_t}\right\|^2 + C_2\sum_{t=2}^{T}\left\|\frac{\alpha_t}{\sqrt{\hat{v}_t}} - \frac{\alpha_{t-1}}{\sqrt{\hat{v}_{t-1}}}\right\|_1 + C_3\sum_{t=2}^{T-1}\left\|\frac{\alpha_t}{\sqrt{\hat{v}_t}} - \frac{\alpha_{t-1}}{\sqrt{\hat{v}_{t-1}}}\right\|^2\right] + C_4$$

$$\leq C_1 H^2/c^2(1+\log T) + C_2 d/c + C_3(d/c)^2 + C_4 \quad (41)$$

Now we lower bound the effective stepsizes, since $\hat{v}_t$ is exponential moving average of $g_t^2$ and $\|g_t\| \leq H$, we have $(\hat{v}_t)_j \leq H^2$, we have

$$\alpha/(\sqrt{\hat{v}_t})_j \geq \frac{1}{H\sqrt{t}}$$

And thus

$$E\left[\sum_{t=1}^{T}\alpha_i\langle\nabla f(x_t),\nabla f(x_t)/\sqrt{\hat{v}_t}\rangle\right] \geq E\left[\sum_{t=1}^{T}\frac{1}{H\sqrt{t}}\|\nabla f(x_t)\|^2\right] \geq \frac{\sqrt{T}}{H}\min_{t\in[T]}E\left[\|\nabla f(x_t)\|^2\right]$$

$$(42)$$

Then by (3), (41) and (42), we have

$$\frac{1}{H}\sqrt{T}\min_{t\in[T]}E\left[\|\nabla f(x_t)\|^2\right] \leq C_1H^2/c^2(1+\log T)+C_2d/c+C_3d/c^2+C_4$$

which is equivalent to

$$\min_{t\in[T]}E\left[\|\nabla f(x_t)\|^2\right]$$
$$\leq \frac{H}{\sqrt{T}}\left(C_1H^2/c^2(1+\log T)+C_2d/c+C_3d/c^2+C_4\right)$$
$$= \frac{1}{\sqrt{T}}\left(Q_1+Q_2\log T\right)$$

One more thing is to verify the assumption $\|\alpha_t m_t/\sqrt{\hat{v}_t}\| \leq G$ in Theorem 3.1, since $\alpha_{t+1}/(\sqrt{\hat{v}_{t+1}})_j \leq \alpha_t/(\sqrt{\hat{v}_t})_j$ and $\alpha_1/(\sqrt{\hat{v}_1})_j \leq 1/c$ in the algorithm, we have $\|\alpha_t m_t/\sqrt{\hat{v}_t}\| \leq \|m_t\|/c \leq H/c$.

This completes the proof.                                                                 **Q.E.D.**

### 6.2.4   PROOF OF COROLLARY 3.2

**Proof.** [Proof of Corollary 3.2]

---
**Algorithm 4. AdaFom**

**(S0).** Define $m_0 = 0$, $\hat{v}_0 = 0$;
For $t = 1, \cdots, T$, do
   **(S1).** $m_t = \beta_{1,t}m_{t-1} + (1-\beta_{1,t})g_t$
   **(S2).** $\hat{v}_t = (1-1/t)\hat{v}_{t-1} + (1/t)g_t^2$
   **(S3).** $x_{t+1} = x_t - \alpha_t m_t/\sqrt{\hat{v}_t}$
End

---

The proof is similar to proof for Corollary 3.1, first let's bound RHS of (3) which is

$$E\left[C_1\sum_{t=1}^{T}\left\|\alpha_t g_t/\sqrt{\hat{v}_t}\right\|^2 + C_2\sum_{t=2}^{T}\left\|\frac{\alpha_t}{\sqrt{\hat{v}_t}}-\frac{\alpha_{t-1}}{\sqrt{\hat{v}_{t-1}}}\right\|_1 + C_3\sum_{t=2}^{T-1}\left\|\frac{\alpha_t}{\sqrt{\hat{v}_t}}-\frac{\alpha_{t-1}}{\sqrt{\hat{v}_{t-1}}}\right\|^2\right] + C_4$$

We recall from Table 1 that in AdaGrad, $\hat{v}_t = \frac{1}{t}\sum_{i=1}^{t}g_i^2$. Thus, when $\alpha_t = 1/\sqrt{t}$, we obtain $\alpha_t/\sqrt{\hat{v}_t} = 1/\sum_{i=1}^{t}g_i^2$. We assume $\min_{j\in[d]}|(g_1)_j| \geq c > 0$, which is equivalent to $\min_{j\in[d]}(\sqrt{\hat{v}_1})_j \geq c > 0$ (a requirement of the AdaGrad). For $C_1$ term we have

$$E\left[\sum_{t=1}^{T}\left\|\alpha_t g_t/\sqrt{\hat{v}_t}\right\|^2\right] = E\left[\sum_{t=1}^{T}\left\|\frac{g_t}{\sqrt{\sum_{i=1}^{t}g_i^2}}\right\|^2\right] = E\left[\sum_{j=1}^{d}\sum_{t=1}^{T}\frac{(g_t)_j^2}{\sum_{i=1}^{t}(g_i)_j^2}\right]$$
$$\leq E\left[\sum_{j=1}^{d}\left(1-\log((g_1)_j^2)+\log\sum_{t=1}^{T}(g_t)_j^2\right)\right] \leq d(1-\log(c^2)+2\log H+\log T)$$

where the third inequality used Lemma 6.9 and the last inequality used $\|g_t\| \leq H$ and $\min_{j\in[d]}|(g_1)_j| \geq c > 0$.

For $C_2$ term we have

$$E\left[\sum_{t=2}^{T}\left\|\frac{\alpha_t}{\sqrt{\hat{v}_t}}-\frac{\alpha_{t-1}}{\sqrt{\hat{v}_{t-1}}}\right\|_1\right]=E\left[\sum_{j=1}^{d}\sum_{t=2}^{T}\left(\frac{1}{\sqrt{\sum_{i=1}^{t-1}(g_i)_j^2}}-\frac{1}{\sqrt{\sum_{i=1}^{t}(g_i)_j^2}}\right)\right]$$

$$=E\left[\sum_{j=1}^{d}\left(\frac{1}{\sqrt{(g_1)_j^2}}-\frac{1}{\sqrt{\sum_{i=1}^{T}(g_i)_j^2}}\right)\right]\le d/c$$

For $C_3$ term we have

$$E\left[\sum_{t=2}^{T-1}\left\|\frac{\alpha_t}{\sqrt{\hat{v}_t}}-\frac{\alpha_{t-1}}{\sqrt{\hat{v}_{t-1}}}\right\|^2\right]$$

$$\le E\left[\frac{1}{c}\sum_{t=2}^{T-1}\left\|\frac{\alpha_t}{\sqrt{\hat{v}_t}}-\frac{\alpha_{t-1}}{\sqrt{\hat{v}_{t-1}}}\right\|_1\right]$$

$$\le d/c^2$$

where the first inequality is due to $|(\alpha_t/\sqrt{\hat{v}_t}-\alpha_{t-1}/\sqrt{\hat{v}_{t-1}})_j|\le 1/c$.

Now we lower bound the effective stepsizes $\alpha_t/(\sqrt{\hat{v}_t})_j$,

$$\frac{\alpha_t}{(\sqrt{\hat{v}_t})_j}=\frac{1}{\sqrt{\sum_{i=1}^{t}(g_i)_j^2}}\ge\frac{1}{H\sqrt{t}},$$

where we recall that $\alpha_t=1/\sqrt{t}$ and $\|g_t\|\le H$. Following the same argument in the proof of Corollary 3.1 and the previously derived upper bounds, we have

$$\frac{\sqrt{T}}{H}\min_{t\in[T]}E\left[\|\nabla f(x_t)\|^2\right]\le C_1 d(1-\log(c^2)+2\log H+\log T)+C_2 d/c+C_3 d/c^2+C_4$$

which yields

$$\min_{t\in[T]}E\left[\|\nabla f(x_t)\|^2\right]$$

$$\le\frac{H}{\sqrt{T}}\left(C_1 d(1-\log(c^2)+2\log H+\log T)+C_2 d/c+C_3 d/c^2+C_4\right)$$

$$=\frac{1}{\sqrt{T}}\left(Q'_1+Q'_2\log T\right)$$

The last thing is to verify the assumption $\|\alpha_t m_t/\sqrt{\hat{v}_t}\|\le G$ in Theorem 3.1, since $\alpha_{t+1}/(\sqrt{\hat{v}_{t+1}})_j\le\alpha_t/(\sqrt{\hat{v}_t})_j$ and $\alpha_1/(\sqrt{\hat{v}_1})_j\le 1/c$ in the algorithm, we have $\|\alpha_t m_t/\sqrt{\hat{v}_t}\|\le\|m_t\|/c\le H/c$.

This completes the proof. **Q.E.D.**

**Lemma 6.9.** *For $a_t\ge 0$ and $\sum_{i=1}^{t}a_i\ne 0$, we have*

$$\sum_{t=1}^{T}\frac{a_t}{\sum_{i=1}^{t}a_i}\le 1-\log a_1+\log\sum_{i=1}^{T}a_i.$$

**Proof.** [Proof of Lemma 6.9] We will prove it by induction. Suppose

$$\sum_{t=1}^{T-1}\frac{a_t}{\sum_{i=1}^{t}a_i}\le 1-\log a_1+\log\sum_{i=1}^{T-1}a_i,$$

we have

$$\sum_{t=1}^{T}\frac{a_t}{\sum_{i=1}^{t}a_i}=\frac{a_T}{\sum_{i=1}^{T}a_i}+\sum_{t=1}^{T-1}\frac{a_t}{\sum_{i=1}^{t}a_i}\le\frac{a_T}{\sum_{i=1}^{T}a_i}+1-\log a_1+\log\sum_{i=1}^{T-1}a_i.$$

Applying the definition of concavity to $\log(x)$, with $f(z) \triangleq \log(z)$, we have $f(z) \leq f(z_0) + f'(z_0)(z - z_0)$, then substitute $z = x - b, z_0 = x$, we have $f(x - b) \leq f(x) + f'(x)(-b)$ which is equivalent to $\log(x) \geq \log(x - b) + b/x$ for $b < x$, using $x = \sum_{i=1}^{T} a_i, b = a_T$, we have

$$\log \sum_{i=1}^{T} a_i \geq \log \sum_{i=1}^{T-1} a_i + \frac{a_T}{\sum_{i=1}^{T} a_i}$$

and then

$$\sum_{t=1}^{T} \frac{a_t}{\sum_{i=1}^{t} a_i} \leq \frac{a_T}{\sum_{i=1}^{T} a_i} + 1 - \log a_1 + \log \sum_{i=1}^{T-1} a_i \leq 1 - \log a_1 + \log \sum_{i=1}^{T} a_i.$$

Now it remains to check first iteration. We have

$$\frac{a_1}{a_1} = 1 \leq 1 - \log(a_1) + \log(a_1) = 1$$

This completes the proof. **Q.E.D.**

