# OpenReview forum: "On the Convergence of A Class of Adam-Type Algorithms  for Non-Convex Optimization"
_ICLR.cc/2019/Conference_

### Official Review · AnonReviewer3 · 2018-11-02
**This paper investigates the convergence condition of Adam-type optimizers in the unconstrained non-convex optimization problems.**

**Rating:** 6
**Confidence:** 3

**Review:**

The main theory points out two scenarios causing Adam-type optimizers to diverge, which extends Reddi et al's results.

The theorem in this paper applies to all Adam-type algorithms, which combine momentum with adaptive learning rates and thus are more general as compared to the recent papers, such as Zhou et al's. The relationship between optimizers' effective step size, step size oscillation and convergence is well demonstrated and is interesting.

Remarks:
1. The main theorem and proof are based on the non-convex settings while the examples to demonstrate the convergence condition are simple convex functions.

2. The message delivered by MNIST experiment is limited, is not clear and is not very relevant to the main part of the paper. It would be better to compare these algorithms in larger deep learning tasks.

Typo:
Page 5, section 3.1: Term A is a generalization of term alpha^2 g^2 (instead of just alpha^2) for SGD.

---

> ### Author Response · Authors · 2018-11-24
> **Response to reviewer 3**
>
> Thank you for your positive review. In the experiment in section 3.2.2, we consider a convex function f = \sum f_i, where f_1 is convex but f_i are nonconvex for i > 1. However, it is easy to extend the above function f to the nonconvex setting if needed. Because the divergence results are essentially caused by function properties around neighborhoods of local minima, we can change the shape of the convex function outside a small region around local minimum to make it non-convex (we will get similar experiment result and our analysis will still work). But we tried to make our examples as simple as possible so that the reason of divergence can be easily understood.
>
> The purpose of the MNIST experiment is just to confirm the performances of different algorithms. To test our algorithms in a larger problem setting, we have added experiments for training CIFARNET on CIFAR-10 in Section 4.

---

### Official Review · AnonReviewer1 · 2018-11-05
**A Comment on the Convergence of A Class of Adam-Type Algorithms for Non-Convex Optimization**

**Rating:** 7
**Confidence:** 2

**Review:**

The work studies the convergence properties of a "Adam-type" class of optimization algorithms used for neural network.
The “Adam-type” class includes the popular algorithms such as Adam, AMSGrad and AdaGrad. Mathematical analysis is conducted to study the convergence of those algorithms in the non-convex setting. The authors derive theorems that guarantee the convergence of Adam-type algorithms under certain conditions to first-order stationary solutions of the non-convex problem, with O(log T /√ T) convergence rate. These conditions for convergence presented in this work is are “tight”, in the sense that violating them can make an algorithm diverge. In addition, these conditions can also be checked in practice to monitor empirical convergence, which gives a positive practical aspect to this work. The authors propose a correction to the Adam algorithm to prevent an option of divergence, and propose a new algorithm called AdaFom accordingly.
Overall this seems like a high-quality work with interesting contribution to the research community. This reviewer is not an expert in theoretical analysis of optimization algorithms, therefore it is hard to assess the true contribution of this work and its comparison to other works in this field.

---

> ### Author Response · Authors · 2018-11-24
> **Response to reviewer 1**
>
> Thank you for your positive comments. Aside from our theoretical analysis, you may find our experiments and discussions in Section 6.1.1 interesting. We compared the performance of Adam, AMSGrad, SGD on a quadratic problem and our finding is that adaptive methods are more robust to choices of stepsizes. This can be beneficial when the structure of the optimization problem is unknown and the best stepsizes are difficult to obtain.

---

### Official Review · AnonReviewer4 · 2018-11-07
**Good paper; waiting for clarification on a few points**

**Rating:** 7
**Confidence:** 3

**Review:**

Summary:

This paper presents a convergence analysis in the non-convex setting for a family of optimization algorithms, which the authors call the "Adam-type". This family incorporates popular existing methods like Adam, AdaGrad and AMSGrad. The analysis relies only on standard assumptions like Lipschitz smoothness and bounded gradients.

Individual Comments/Questions:

- In Table 1, the characterization of Adam ignores the fact that, in practice, Adam adds a positive epsilon to the $\hat{v}_t$ in the denominator. I would like the authors to at least comment on that in the paper. I assume that AdaFom and AMSGrad also need such an epsilon in practice. Could the author comment on whether (and how) this would affect their analysis of those methods? In particular, in Theorem 3.1, can we really assume the term $\Vert \alpha_t m_t / \sqrt{\hat{v}_t} \Vert$ to be bounded by a constant without such an epsilon?

- In the first bullet point in Section 3.1, the authors relate the term

$$ \sum_t \Vert \alpha_t g_t / \sqrt{\hat{v}_t} \Vert^2 $$ (*)

to the term $\sum_t \alpha_t^2$ in the analysis of SGD. I don't think this is a fair analogy. The effective step size of Adam-type methods is $\alpha_t / \sqrt{\hat{v}_t}$, while the aformentioned term also contains the magnitude of the stochastic gradient $g_t$. So, while the SGD analysis only poses a condition on the step sizes, bounding (1) also poses a condition on the magnitude of the stochastic gradient.

- In the experiments of Section 3.2.1, the authors use a step size of 0.01 for SGD (which really is gradient descent, since this is a non-stochastic problem). Existing theory tells us that GD only converges for step sizes smaller than 2/L where L is the Lipschitz constant of the gradient, which is L=200 in this example. So this is literally setting the method up for failure and I don't really see any merit in that experiment.

- The experiments in Section 4 are of course very limited, but this paper makes a significant theoretical contribution, so I don't really see the need for extensive experiments.

- To my knowledge, under similar assumptions, plain SGD has been show to converge at a rate of O(1/sqrt(T)). The convergence analysis presented here has an additional log(T) factor, so it is not really suitable to explain any possible benefits of these adaptive methods over SGD. This is totally fine in and of itself; after all the analysis of theses methods is hard and this is a great first step. The issue I have is that this is not mentioned in the paper at all.

Originality:

To the best of my knowledge, the convergence analysis of the Adam-type methods (including established methods AdaGrad, RMSprop, Adam, AMSGrad) in the _non-convex_ setting is a novel, original contribution. The authors also propose a new algorithm, AdaFom. This exact algorithm is proposed in [1], which was uploaded to arXiv before the ICLR deadline. However, this can be considered concurrent work.

Significance:

The convergence properties of popular optimization methods in machine learning (e.g., Adam) are generally very poorly understood in "realistic" settings. The analysis presented in this paper is an important step to better theoretical understanding of these methods which, in my opinion, is highly significant.

Correctness:

This was a short-notice emergency review and I did not check any of the proofs in the appendix. I will try to verify at least parts of the proofs in the coming days.

Conclusion:

This is an original paper making a significant theoretical contribution. I can't comment on the correctness of the mathematical analysis (yet). I'm cautiously recommending acceptance for now, but would be willing to upgrade my rating if the authors respond to my comments/questions.


[1] Zou and Shen. On the Convergence of Weighted AdaGrad with Momentum for Training Deep Neural Networks. https://arxiv.org/abs/1808.03408.

--------------------------------
Update
--------------------------------

The authors have provided a detailed response to my concerns and have fixed many of them in their revised version. I verified parts of the proofs in the appendix (Theorem 3.1 and its Corollaries). I congratulate the authors on their work and recommend acceptance!

---

> ### Author Response · Authors · 2018-11-24
> **Response to reviewer 4**
>
> Thank you for your valuable feedback. We respond your comments and questions point by point.
>
> Adding $\epsilon$:
> For  Generalized Adam (Algorithm 1), adding $\epsilon$ to $\hat{v}_t$ does not affect our convergence analysis. Our Theorem 3.1 still holds since in Algorithm 1, $\hat{v}_t$ takes a very general form that can cover the cases where $\hat{v}_t$ is lower bounded by manually adding an $\epsilon$.
> Then the real question is for specific algorithms, can the term $\Vert \alpha_t m_t / \sqrt{\hat{v}_t} \Vert$ in Theorem 3.1 be upper bounded by a constant?
>
> First, the reviewer is correct that adding epsilon helps in this regard because it lower-bounds $\hat{v}_t$ by epsilon and $\Vert \alpha_t m_t / \sqrt{\hat{v}_t} \Vert$ can be easily upper bounded. In this case,our theory still holds. We have clarified this point after Theorem 3.1.
>
> However, for specific algorithms AdaFom and AMSGrad, we actually does not need to add $\epsilon$ to ensure convergence. In particular, by assuming $\|g_t\|$ is upper bounded as in Assumption A2, the upper boundedness of $\Vert \alpha_t m_t / \sqrt{\hat{v}_t} \Vert$  is automatically satisfied. The upper boundedness is verified at the end of Section 6.2.3 and Section 6.2.4. We added a comment after Theorem 3.1 that the upper boundedness of the term is automatically satisfied by AdaFom and AMSGrad according to your suggestion.
>
> Analogy to SGD:
> You are right. Rigorously, $\sum_{t} \| \alpha_t m_t / \sqrt{\hat{v}_t} \|^2$ should be an generalization of $\sum_{t} \| \alpha_t g_t \|^2$ in SGD. We made the analogy because under the assumption that $\|g_t\| \leq G$, $\sum_{t} \| \alpha_t m_t / \sqrt{\hat{v}_t} \|$ is upper bounded by  $G^2 \sum_{t}  \| \alpha_t / \sqrt{\hat{v}_t} \|^2 $. The latter transfers to $\sum_{t}\alpha_t^2 $  when ignoring other constants (G and dimension d). We have changed $\sum_{t}\alpha_t^2$ to $\sum_{t} \| \alpha_t g_t \|^2$ in the paper to avoid confusion.
>
>
> Experiments in Section 3.2.1:
> The purpose of the experiment in Section 3.2.1 is to show that Adam-type algorithm may diverge when term A in (5) grows too fast. Both the divergence of Adam and SGD serve as support to the aforementioned claim. For SGD, we intentionally set up an experiment to make it diverge and observe whether the growth of term A in (5) agrees with our theory. This is only to verify our theory and the purpose is not to provide new insights about SGD (since SGD is well-studied ).
>
> On the contrary, the divergence of Adam in the experiment is the interesting part and it provides some new insights. The experiment says that even for batch versions, Adam is not guaranteed to converge to a stationary point using a small constant stepsize (this is different from batch GD which is guaranteed to converge using a  stepsize smaller than 2/L). This is a new discovery in our experiment.  The  message we want to convey is the following: the convergence requirement of limiting the growth rate of term A to be slower than that of accumulation of effective stepsizes in (5) is not an artifact, or being too restrictive; because  SGD and Adam can diverge even when term A grows with the same speed as the accumulation of effective stepsizes.
>
> Experiments in Section 4:
> Thank you for your support. To make our experiments more convincing, we have added experiments on larger problems (training CIFARNET on CIFAR-10) in the revised paper (Section 4).
>
>
> Possible benefits:
> Thank you for pointing out this possible confusion. In our new version, we explicitly point out that the possible benefits of adaptive methods may not lie in the worst case convergence rates, but can be explained by other factors such as robustness to choices of stepsizes (the last paragraph in Section 3.3).
>
> As mentioned, we are not explaining the benefits by claiming faster worst-case convergence rate. The main benefit we are showing is the robustness to the choice of stepsizes compared to SGD. You may find experiments in Section 6.1.1 interesting where we compare performance of SGD and adaptive methods under different choices of stepsizes. The message obtained from the experiments is that, adaptive methods may converge or converge faster for a larger range of stepsizes compared to SGD. Therefore adaptive methods may perform much better on average if we choose stepsizes randomly (which is usually the case in practice since it is time consuming to test all possible values of stepsizes). In the future, we will try to show the possible benefits of certain adaptive methods in a more rigorous way .
>
> We have also commented on the presence of the additional log term in convergence rates at the end of Section 3 in the revised manuscript.

---

> > ### Author Response · Authors · 2018-11-24
> > **Response to reviewer 4 continued**
> >
> > Algorithm in [1]:
> > Thank you for for emphasizing this highly related work. Thanks to explanation provided in new version of [1], we found AdaFom is slightly different from the algorithm (AdaHB) in [1]. As mentioned in Section 4 of [1], $m_t$ in AdaFom is an exponential moving average of $g_t$ while $m_t$ in AdaHB in [1] is a exponential moving average of $\alpha_t g_t /\hat{v}_t$.
> >
> > Again, thank you for your positive feedback and timely review.

---

> > > ### Comment · AnonReviewer4 · 2018-11-27
> > > **Thanks for the detailed reply**
> > >
> > > I thank the authors for their detailed reply.
> > >
> > > The revised version contains clarifications and explanatory comments on the "cumulative step sizes" (and how that quantity relates to SGD) as well as the additional log(T) factor in the convergence rate compared to SGD. I think this greatly improves the clarity of the paper.
> > >
> > > Regarding the epsilon offset for some of the adaptive methods: I had missed the relevant parts of the proofs of Corollary 3.1 and 3.2; thanks for pointing them out. However, the fact that this does not need an epsilon relies on a hidden assumption that the initialization point is such that no gradient coordinate is zero. I would ask the authors to clearly state that assumption in the text of the Corollaries.
> > >
> > > Also, I stand by my point that the analysis would be much more interesting if it would incorporate any possible effects of the epsilon since, to my knowledge, this is an practically important aspect of these Adam-style methods. (Alternatively, the authors might want to explain why they think that such an epsilon would not considerably affect the theoretical results.)

---

> > > > ### Author Response · Authors · 2018-11-29
> > > > **Thank you for standing by your point**
> > > >
> > > > Assumption that no gradient coordinate is 0 at initial point:
> > > >
> > > > We will explicitly state this assumption in the next version. Specifically, we will add “Assume that $\|(g_{1})_i| \geq c,  \forall i$” in the Corollary 3.1 and 3.2.
> > > >
> > > >
> > > > Analysis after adding $\epsilon$:
> > > >
> > > > Your comment on $\epsilon$ is very helpful! We will add more discussion on effect of $\epsilon$ in our next version (or a future arxiv version if adding a few corollaries will be considered too much).
> > > >
> > > > After going through the proof with $\epsilon$ added, we find that adding a proper $\epsilon$ can indeed help with worst-case convergence rate in our analysis. Meanwhile, with $\epsilon$ added, we do not need to assume that no gradient coordinate is zero at the first iteration.
> > > >
> > > > Since we are not able to modify the paper at this stage, we go through what will happen in the proof of Corollary 3.1 here (it is just a simple modification of the original proof).
> > > >
> > > > We consider adding $\epsilon$ as replacing S4 in Algorithm 3 (AMSGrad) to $x_{t+1} = x_t - \alpha_t m_t / (\sqrt{\hat{v}_t} + \epsilon)$. This resulting algorithm is also a special case of Algorithm 1 because $(\sqrt{\hat{v}_t} + \epsilon)$ is still a function of all past gradient (Theorem 3.1 still holds). Then we can use (5) to analyze the convergence of the new Algorithm 3. Briefly speaking, the analysis requires only a few modifications in the proof of Corollary 3.1 ($(\hat{v_1})_j \geq c$ replaced by ($(\hat{v_1})_j + \epsilon \geq \epsilon$ and $(\hat{v_t})_j \leq H$ replaced by $(\hat{v_t})_j + \epsilon \leq H + \epsilon)$).
> > > >
> > > > Now we provide a detailed discussion on modifications in the proof of Corollary 3.1 if $\epsilon$ is added. Our following discussion is based on Section 6.2.3 in the paper. As the reviewer may already noticed, our proof of Corollary 3.1 relies on upper bounding Term A and Term B in (5) of Theorem 3.1 and finding G in the assumption $\| \alpha_t m_t / \sqrt{\hat{v}_t} \| \leq G$. With $\epsilon$ added in the denominator, we can easily have $\| \alpha_t m_t / \sqrt{\hat{v}_t} \| \leq H/\epsilon$ because we have $\|m_t\| \leq H$ (due to $\|g_t\| \leq H$).
> > > >
> > > > We now discussed how $\epsilon$ affects the rest of the proof.
> > > >
> > > > Above the first unnumbered inequality in page 27, we have assumed that $(\hat{v}_1)_j \geq c > 0$. With $\epsilon$ added, we have $(\hat{v}_1)_j + \epsilon \geq \epsilon$. Thus we can replace $c$ in the remaining proof to $\epsilon$ and allow $(\hat{v}_1)_j = 0$ (which removes the assumption that the first gradient is non-zero at every coordinate).
> > > >
> > > > The rest of the proof will easily go through by replacing $\hat{v}_t$ to $\hat{v}_t+ \epsilon$ until (41). ( $\hat{v}_t+ \epsilon$ is still monotonically increasing since $\hat{v}_t$ is, this enables the telescoping sum in (40)).
> > > >
> > > > Notice that we have used $(\sqrt{\hat{v}_t})_j \leq H$ below (41), with $\epsilon$ added, we replace it by $(\sqrt{\hat{v}_t})_j + \epsilon \leq  H + \epsilon$. Then the $1/H$ will be replaced to $1/(H+\epsilon)$ from the unnumbered inequality below (41) to the inequality below (42). Then we can substitute $G=H/\epsilon$ and expression of $C_1, C_2, C_3, C_4$ (at the end of Section 6.2.2) into the final bound at the end of the proof.
> > > >
> > > > In the end, the RHS of the last inequality before the end of the proof has the form $A/\epsilon^2 + B/\epsilon + C \epsilon + D$ with $A,B,C,D$ being numbers independent of $\epsilon$. Thus a proper $\epsilon$ can be chosen to minimize the upper bound.
> > > >
> > > > It is easy to see that the optimal $\epsilon$ is neither $\infty$ nor 0 but something in between. Intuitively, this is because when $\epsilon$ is very large, the effective stepsizes will be very small and the algorithm will not make fast progress. When $\epsilon$ is very small, the algorithm may move unpredictably due to very large effective stepsizes at the early stage of optimization. Again, the theoretical bound agrees with what people observe in practice.
> > > >
> > > > The analysis of AdaFom at the presence of $\epsilon$ will be similar (even simpler than the current proof of Corollary 3.2 because it is easier to bound Term A in (5)).
> > > > The main reason for ignoring $\epsilon$ in our original version is its simplicity in convergence analysis. However, based on the discussion with the reviewer, we think it is very interesting to see how $\epsilon$ affects the performance of AMSGrad and AdaFom. Again, we will add more discussion on adding $\epsilon$ in our next version . We sincerely hope that our response have addressed the reviewer’s concern.

---

> > > > > ### Comment · AnonReviewer4 · 2018-11-29
> > > > > **Final response**
> > > > >
> > > > > Without having followed all the details in the above response, the outlined analysis including the $\epsilon$ sounds very interesting. I would encourage the authors to include this in a future version of the paper.
> > > > >
> > > > > In the meantime, I verified some of the proofs in the appendix, specifically those of Theorem 3.1 as well as the Corollaries.
> > > > >
> > > > > Overall, I think this is a good paper.  I will update my review and increase my rating shortly.

---

### Author Response · Authors · 2018-11-24
**General response to all reviewers:**

We thank all reviewers for the positive comments on our paper. Regarding the reviewer’s specific questions, we have carefully addressed each of them, and further revised our paper and presentation. We have also enriched our experiments by including the example of training CIFARNET model on CIFAR-10 dataset.

---

### Meta-Review · Area_Chair1 · 2018-12-14

**Confidence:** 4
**Recommendation:** Accept (Poster)

**Metareview:**

This paper analysis the convergence properties of a family of 'Adam-Type' optimization algorithms, such as Adam, Amsgrad and AdaGrad, in the non-convex setting. The paper provides of the first comprehensive analyses of such algorithms in the non-convex setting. In addition, the results can help practitioners with monitoring convergence in experiments. Since Adam is a widely used method, the results have a potentially large impact.

The reviewers agree that the paper is well-written, provides interesting new insights, and that is results are of sufficient interest to the ICLR community to be worthy of publication.